# Mapping Robusta coffee (*Coffea canephora*) cropping systems in Uganda: A two-step pixel and sub-pixel based approach with Sentinel-2 data

Getachew Kebede[1,2*], Bester Tawona Mudereri[1,3], Onisimo Mutanga[2], Tobias Landmann[1], John Odindi[2], Natacha Motisi[1,4], Fabrice Pinard[1,4], Henri E. Z. Tonnang[2], Elfatih M. Abdel-Rahman[1,2]

**1** International Centre of Insect Physiology and Ecology (icipe), Nairobi, Kenya, **2** School of Agriculture and Science, University of KwaZulu-Natal, Pietermaritzburg, South Africa, **3** International Potato Centre (CIP), Kigali, Rwanda, **4** Centre de Coopération Internationale en Recherche Agronomique pour le Développement, (CIRAD), UMR PHIM, Nairobi, Kenya

\* garagaw@icipe.org

## Abstract

Coffee is a highly valued commodity and a widely consumed beverage, playing an important role in global trade. However, coffee farming landscapes are increasing transitioning into smaller-scale agricultural setups. This transformation highlights the critical need for accurate classification and mapping of coffee cropping systems (CS), especially in countries like Uganda, where dense vegetation and complex terrain present substantial challenges to traditional land survey methods. Moreover, understanding the spatial distribution of Robusta coffee (*Coffea canephora*) CS is essential for developing site-specific management strategies, guiding extension services, and informing evidence-based policy decisions. To address this gap, the present study aimed to enhance the discrimination and mapping capabilities of Robusta coffee CS at a sub-pixel scale using a two-step classification approach and multi-date Sentinel-2 (S2) data. In the first step, the random forest (RF) classification algorithm was used to map the major land use and land cover (LULC) classes in Google Earth Engine platform. Then, the Robusta coffee cropland class was masked, and a sub-pixel multiple endmember spectral mixture analysis (MESMA) was employed to discriminate Robusta coffee CS using three endmembers (EMs) obtained from *in-situ* hyperspectral data collected in 2023 from: (i) Robusta coffee with agroforestry, (ii) Robusta coffee with banana, and (iii) Robusta coffee under full sun. The result showed 93.5% overall accuracy for the major LULC and 89.9% for all Robusta coffee CS classes, disentangled as follows: 91.3% accuracy for Robusta coffee with agroforestry, 88.5% for Robusta coffee with banana, and 91.2% for Robusta coffee under full sun. Moreover, the MESMA sub-pixel algorithm demonstrates credible performance in discriminating the Robusta coffee CS within each S2 pixel at the heterogeneous

**Data availability statement:** The data used in this study are publicly available on the icipe data portal: https://dmmg.icipe.org/dataportal/dataset/mapping-robusta-coffee-coffea-canephora-cropping-systems-in-uganda.

**Funding:** Getachew Kebede was supported by the In-Region Postgraduate Scholarship from the German Academic Exchange Service (DAAD). The authors sincerely appreciate the financial support provided by the European Union (EU) for the project "Robusta coffee agroforestry to adapt and mitigate climate change in Uganda" (GCCA+-Global Climate Change Alliance & DESIRA, Project/Grant No. FOOD/2021/427-759). Additional funding was received from the Swedish International Development Cooperation Agency (Sida), the Swiss Agency for Development and Cooperation (SDC), the Australian Centre for International Agricultural Research (ACIAR), the Norwegian Agency for Development Cooperation (Norad), the Federal Democratic Republic of Ethiopia, and the Government of the Republic of Kenya. The project also benefited from funding by the Centre de Coopération Internationale en Recherche Agronomique pour le Développement (CIRAD). Internal support was also provided by the authors' host institutions through logistical, technical, and administrative assistance. There was no additional external funding received for this study.

**Competing interests:** The authors have declared that no competing interests exist.

landscape in the study area. The findings of this study can inform site-specific interventions (e.g., pest management, fertilizer application, etc.) tailored to the type of CS.

## 1. Introduction

Uganda is Africa's second-largest coffee producer after Ethiopia, with over 1.5 million small-scale coffee farmers accounting for 10% of global coffee production [1]. These farmers cultivate both Arabica (*Coffea arabica* L. - 23% of the total production) and Robusta (*Coffea canephora* −77% of the total production) coffee on plots smaller than 0.5 ha, often intercropped with bananas and various agroforestry species [2]. In the recent past, Uganda has faced challenges arising from climate change, that potentially require the replanting or regenerating of older coffee trees. This may explain why Uganda's coffee industry underperforms compared to the other major producers like Vietnam and Brazil. The Uganda coffee technology adoption, for instance, are hindered by so many factors as opposed to major producers [3]. To address these constraints, Okia & Buyinza [4] recommend on-farm participatory trials to evaluate shade tree compatibility and determine the best spatial and temporal configurations for coffee cropping systems (CS) in the country. Also, researchers promote implementing a cluster approach, tailored to the specific type of coffee CS, where farmer groups collaborate to seek government intervention or support. Furthermore, the Uganda Coffee Development Authority advocates for strategies such as promoting commercial cultivation, replanting, adapting to climate change, and accessing new markets.

Efforts to map coffee CS using remote sensing approaches in Uganda face significant challenges due to dense vegetation and varied terrain. These difficulties are particularly pronounced in coffee CS with dense shade trees [5]. Issues such as topographical variations, complex CS structures (e.g., coffee farm or patch layout, species heterogeneity within and between coffee farms), and the presence of diverse flora and shade trees further complicate accurate coffee CS mapping. Whereas a successful land use and land cover (LULC) classification has been achieved in full sun coffee CS, mapping various coffee CS at a finer landscape scale remain a challenge [6,7]. Hence, some studies have used various data sets, such as high-resolution optical images, spectral indices, and temperature data to enhance the accuracy of classifying coffee agroforestry systems [8,9].

Understanding the spatial distribution of Robusta coffee CS is essential for developing site-specific management strategies, guiding extension services, and informing policy decisions [10,11]. Accurate maps of coffee CS can support targeted pest and disease monitoring [11], facilitate climate adaptation planning [12], and help optimize resource allocation by identifying the most vulnerable or high-potential areas [10]. Despite this importance, few studies have focused on mapping coffee CS at fine scales in heterogeneous landscapes [13]. This study bridges that gap by applying advanced spectral unmixing techniques to classify and map diverse Robusta coffee CS using remote sensing data and advanced machine learning classification algorithms [14].

In the typical small-scale agroecological systems in Africa, inter and intra coffee CS spectral responses are often due to landscape heterogeneity, fragmentation, and intricate crop cycles [15]. Specifically, the subtle spectral characteristics of coffee CS might be uncovered using pixel-based classification methods and image data of medium or high spatial resolution, such as Landsat (30 m) and Sentinel-2 (S2) (10 m). Hence, studies suggest the use of sub-pixel classification methods, such as spectral mixture analysis (SMA) [14], which can effectively decompose mixed pixels with multiple classes and produce fairly acceptable mapping accuracy. Different methods of SMA are successfully used in diverse studies, including simple spectral mixture analysis: sSMA [14], Monte Carlo spectral mixture model: AutoMCU [16], Bayesian spectral mixture analysis: BSMA [17], linear mixture analysis: LAM [18], and multiple endmember spectral mixture analysis: MESMA [19]. These methodologies are implemented by: (1) determining the number of pure spectra known as endmembers (EMs) present in image data, (2) discerning the biophysical characteristics of each EM within a pixel, and (3) estimating the fractional abundances of each EM in a pixel [14,20].

The MESMA algorithm is the most popular sub-pixel classification method due to its low error rates [21]. Furthermore, unlike the spectral mixture analysis, which uses a fixed set of endmembers for all pixels, MESMA allows EM combinations to vary for each pixel, making it more adaptable to spatially heterogeneous surfaces [19]. While deep learning methods can also capture complex spectral–spatial relationships, they typically require large datasets and high computational resources. MESMA provides a more transparent, data-efficient, and interpretable approach for mapping mixed LULC types, especially in areas with limited training data or where physical understanding of spectral behavior is essential [14]. The MESMA has been used in, for instance, a range of fields that include the analysis of urban settings [22], the delineation of fire-affected areas [23,24], discrimination of plant species [19], mapping marshland [25], detection of weeds in maize crop [26] and mapping agro-ecosystems [27].

This study evaluates the performance of MESMA in mapping three distinct Robusta coffee CS using Sentinel-2 imagery in Uganda. We first classified the study landscape into five major LULC classes to distinguish Robusta coffee cropland from other major LULC types. After isolating the Robusta coffee cropland, we applied MESMA exclusively within the Robusta coffee cropland class to further discriminate and map the three Robusta coffee CS at a sub-pixel scale.

## 2. Materials and methods

### 2.1. Materials

**2.1.1. Study site.** This study was conducted in the main Robusta coffee growing-regions in central, eastern, western, and southwestern Uganda, encompassing 94,076 km$^2$. The study site spans an altitude range of 612–5239 meters above sea level, with coordinates at latitudes 1° 32'N to 1° 21' S and longitudes 29° 31' E to 34° 27' E (Fig 1). The area experiences a tropical climate characterized by an annual bimodal rainfall pattern, with temperatures ranging from 18 to 22 °C. The annual precipitation varies between 750 and 1500 mm [1]. The study area is divided into 34 districts with the main Ugandan Robusta coffee growing-regions and mainly dominated by Robusta coffee cropland [1]. Agricultural practices in the study area are primarily subsistence and small-scale, featuring diverse agricultural systems such as agroforestry with timber and fruit trees. Additionally, intercropping with horticultural crops like banana, cassava, and yam is commonly integrated into the Robusta coffee CS. The main challenges to Robusta coffee production in this region include climate change, pests, and diseases.

**2.1.2. Field data collection.** Field surveys were conducted between January 1$^{st}$ and February 28$^{th}$, 2023, to collect ground polygon data on Robusta coffee cropland (n = 287). Field data were collected using an open data kit (ODK) application installed on a smartphone, which captured location information with an accuracy of ±3 m. Field data collection was carried out across the study sites, facilitated and permitted by the Uganda National Coffee Research Institute (NaCORI) and the National Forestry Resources Research Institute (NaFORRI), which are the national entities responsible for overseeing coffee and forest research in Uganda. These entities were also part of the Robusta project consortium, under which this study was conducted. To reduce the potential sampling bias during LULC Robusta coffee cropland

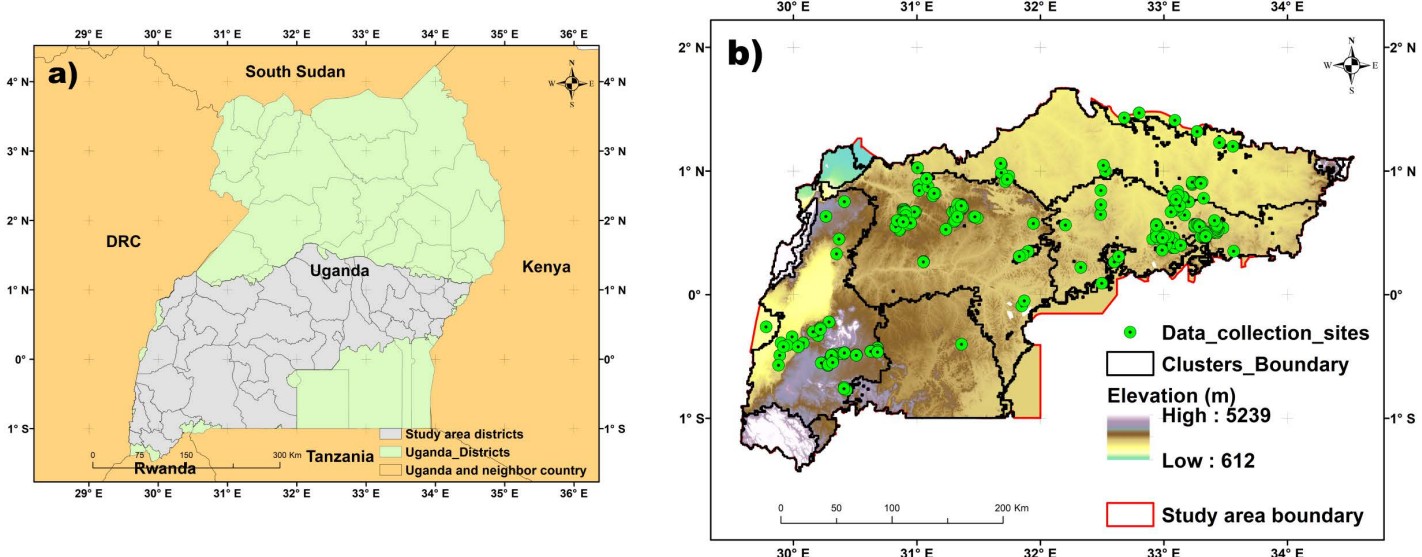

**Fig 1. Location of the main Robusta growing-regions in Uganda and the distribution of Robusta coffee cropping systems (CS) and major land use and land cover (LULC) sample points (n = 1,129), overlaid on a 30 m resolution digital elevation model (DEM) sourced from the United States Geological Survey (USGS).**

reference data collection, we employed a stratified random sampling approach. Specifically, we divided the landscape into eight clusters based on several landscape and climate indicators, including elevation, slope, rainfall, temperature, and soil properties using a k-means cluster analysis (Table 1).

We then randomly collected ground-truth points for Robusta coffee cropland and major LULC as polygons (i.e., regions of interest: ROIs). The major LULC included, natural forests, mixed agricultural and grassland areas (which encompass commercial farms, subsistence farms, grazing lands, and areas with trees and shrubs), built-up areas, Robusta coffee cropland, and water bodies as described in [26,28]. Additionally, we collected ROI for each major LULC class on Google Earth Platform® with a Landsat 8 background image captured on the 15th of January, 2022. The image had spatial resolution of 30 m x 30 m [29]. The number of pixels for ROIs was calculated based on S2 spatial resolution (10 m x 10 m) using Orfeo ToolBox (OTB) in quantum geographic information system [30]. To ensure representative number of pixels used in the classification experiment, a scaling factor of 0.189 was applied to each major LULC ROI and Robusta coffee cropland sample. This approach allowed for proportionally balanced representation of pixel samples across ROIs of varying sizes. As a result, a total of 3,000 pixels were selected for all major LULC classes combined. A summary of the reference sample pixels used is provided in (Table 2). The representative pixel samples were then divided into 70% (2100) for training and 30% (900) for validating the classification model.

**Table 1. Summary of datasets, spatial resolution, and acquisition date used for cluster analysis and Sentinel-2 (S2) imagery-based major land use and land cover (LULC) classification.**

| Dataset name | Source | Spatial resolution | Date used |
|---|---|---|---|
| **Elevation and Slope** | USGS SRTM (Shuttle Radar Topography Mission) | 30 meters | 2022 |
| **Rainfall and Temperature** | CHIRPS (Climate Hazards Group InfraRed Precipitation with Station data) | ~5 kilometers | 2010-2022 |
| **Soil properties** | AfSIS (Africa Soil Information Service – Grid Soil Database) | 250 meters | 2020 |
| **Sentinel-2 (S2)** | Copernicus Programme (European Space Agency) | 10–60 meters (band-dependent) | 2023 |

**Table 2. Description of major land use and land cover (LULC) in the study area, including the number of regions of interest (ROI), number of representative pixel samples, and scaling factor.**

| Major LULC category | Number of pixels for ROI | Representative pixel sample | Scaling factor | Descriptions |
|---|---|---|---|---|
| Natural forest | 2,048 | 340 | 0.189 | Well-stocked forest stands dominated by native species, including primary and mature secondary forests with minimal disturbance. |
| Mixed agricultural and grassland | 11,022 | 2,084 | 0.189 | Expansive commercial farms, typically monoculture, alongside subsistence farms, as well as rangelands, grazing areas, and grounds with trees and shrubs. |
| Built-up areas | 288 | 54 | 0.189 | These are artificial surfaces, including constructed urban areas. |
| Robusta coffee cropland | 1,800 | 340 | 0.189 | Robusta coffee plantations are intercropped with banana, agroforestry trees, mostly timber, and fruit trees, and exposed to direct sunlight. |
| Water bodies | 360 | 68 | 0.189 | Open water bodies include lakes and rivers |
| Total | 15,870 | 3,000 | | |

To address potential concerns regarding spatial autocorrelation between training and validation samples, we included a map (Fig 2) illustrating the distribution of polygon used for training and validating the random forest (RF) classification algorithm. These polygons are distributed randomly across the eight clusters. The digitized polygons area 0.27 to 50.7 hectares. The majority of polygons fall within the 0.27–12 ha range, with a mean size of 5.6 ha. Some polygons are close to each others, but since we randomly selected the training and validation samples, we expect minimum or no autocorrelation among these samples.

The sample polygons were organized at a point pixel scale in QGIS [30], retrieved in the GEE, and their reflectance values were extracted at the S2 spectral (Blue, Red, Green, Red-edge1, Red-edge2, Red-edge3, near-infrared (NIR), and narrow-band near-infrared (NIRn)) bands of a 10-m spatial resolution.

## 2.2. Methods

The proposed methodology employed a two-step classification approach for mapping Robusta coffee CS. In the first step, a multi-date S2 composite was utilized to map the major LULC in the study area. This is achieved using the RF [31]

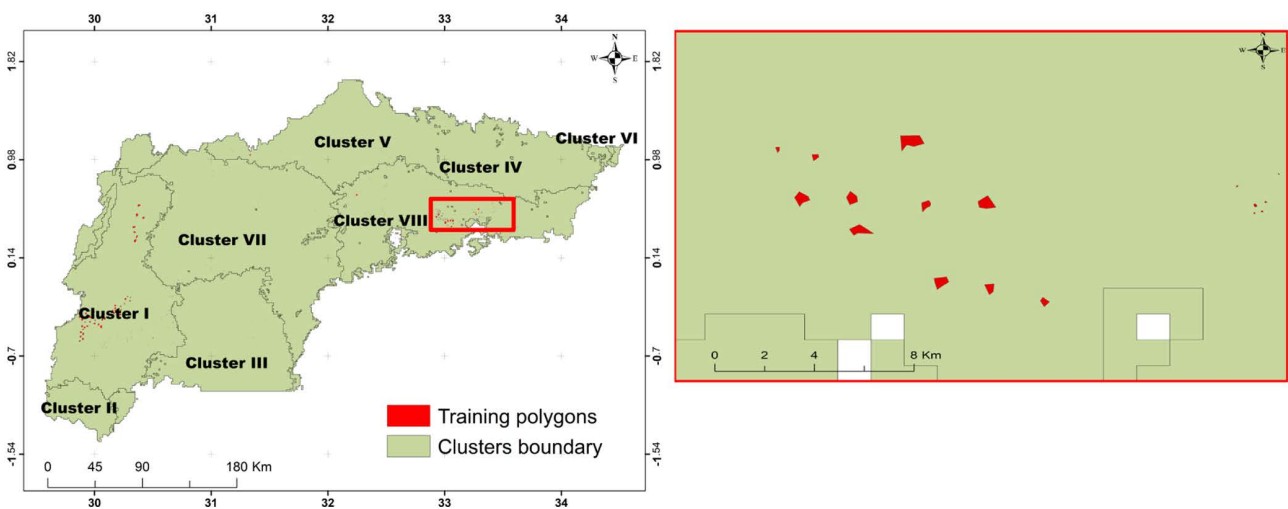

**Fig 2. Distribution of digitized polygons used for training and validation across the eight clusters.** a), and b) zoomed-in view of the distribution and size of polygons within the highlighted red box.

classifier in a semi-automatic approach within the Google Earth Engine (GEE). The second step involved spectral unmixing analysis of the derived Robusta coffee into three CS using MESMA and a dry season S2 median composite image generated from all available images acquired between January 1st and February 28th, 2023 in R-software [32]. The overall workflow of the classification approach utilized in this study is illustrated in Fig 3, with detailed explanations of each step provided in the subsequent sub-sections. To ensure consistency, all spatial data layers and maps were processed and visualized using World Geodetic System (WGS) 84 coordinate system in decimal degrees.

**2.2.1. Sentinel-2 (S2) data compositing and classification in the Google Earth Engine (GEE).** The study employed GEE to map the major LULC. The GEE offers several advantages, including ease of use, access to a comprehensive library of global remotely sensed datasets, powerful cloud-based computing power, and the ability to process large-scale imagery without being constrained by the user local computer memory. This is particularly beneficial when working with high-resolution or global-scale data. A multi-date S2 image composite covering the period from 01/01/2022–31/12/2022 was used in this study. All image processing and classification steps were carried out in the GEE platform. The reflectance data from eight S2 tiles, initially provided as top-of-atmosphere reflectance, were retrieved and used in the major LULC classification experiment. These images underwent a sequence of processing steps,

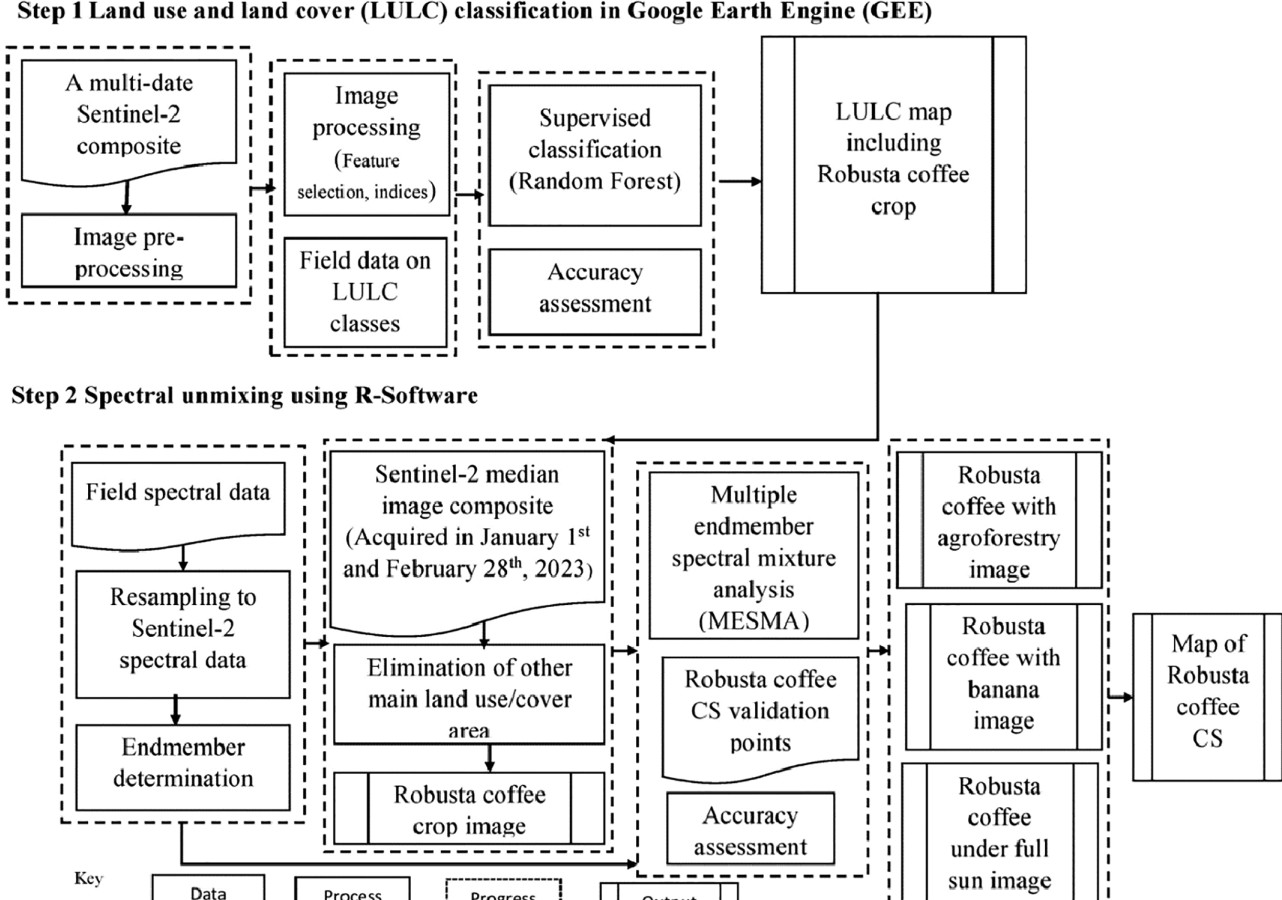

**Fig 3. The overall process of the two-step approach for mapping the major land use and land cover (LULC) and Robusta coffee cropping systems (CS).**

including cloud filtering, atmospheric correction, and normalization for seasonal illumination effects. Median compositing was employed to reduce the influence of redundant or invalid pixels caused by clouds, shadows, or other atmospheric disturbances [33]. The resulting composite was resampled to a 10 m spatial resolution using bilinear interpolation. Removing contaminated pixels is essential to minimize spectral errors induced by atmospheric conditions and ensure more reliable estimation of biophysical parameters [34].

The median compositing method was selected for the classification of both major LULC classes and Robusta coffee CS due to its computational efficiency and superior technical performance compared to other compositing techniques, such as maximum ratio, annual greenness pixel, best pixel (based on the distance to the nearest cloud), and seasonal greenest pixel [33]. By applying the GEE 'median()' function, this method effectively filtered out cloud-contaminated (high-value) and shadow-contaminated (low-values) pixels across all bands, capturing seasonal variability while retaining the most representative pixel values over the entire year [33,35]. Additionally, spatial resampling was performed to standardize all S2 bands to a uniform 10-m spatial resolution.

Five vegetation indices (VIs) were calculated (Table 3) and integrated with S2 spectral bands to support the classification of major LULC in the study area. These indices included chlorophyll vegetation index (CVI), modified simple ratio for vegetation (MSR), normalized difference vegetation index (NDVI), a combination of the modified chlorophyll absorption in reflectance index (MCARI) and optimized soil-adjusted vegetation index (OSAVI), and the Beison Datt vegetation index (DATT). The description, significance and proven effectiveness of these indices for vegetation-based analysis were well documented in previous studies [29,39–42].

We used the RF classification algorithm [31] to map the major LULC classes in the study area. The RF algorithm is renowned for its effectiveness in various remote sensing studies, particularly in tasks such as cropland mapping [43]. The algorithm is an ensemble machine learning method that constructs multiple decision trees (ntree) for classification and regression. Each decision tree is developed using a bootstrap sample, known as "in-bag" data, which constitutes two-thirds of the original dataset. Random subsets of variables (mtry) are employed to split nodes in these decision trees, with the default mtry value calculated as the square root of the total number of variables [44]. Reducing the number of variables utilized in a split not only lowers the algorithm's computing complexity but also reduces the correlation between the ntree. The RF uses a majority voting to label the different classes in the experiment and produce consistent and reliable predictions. For this study, the default settings of ntree (500 trees) and mtry [3] were applied for a pixel-based RF classification and mapping of major LULC classes. It is reported that the performance of RF classifier based on the default ntree and mtry is not significantly different when these two parameters are tuned and optimized [31,44]. The major LULC classes were removed, except the Robusta coffee cropland class, which was retained to generate a mask for further analysis.

Table 3.  Description of the vegetation indices used in major land use and land cover (LULC) classification. The numerical values in the vegetation indices formulas represent central wavelengths (in nanometers), derived from in situ leaf-level hyperspectral reflectance measurements. R = Reflectance.

| Vegetation Index | Equation | Significance | Reference |
|---|---|---|---|
| **Chlorophyll vegetation index (CVI)** | R842/ (R665 - R560) | Indicates chlorophyll content and plant health | [36] |
| **Modified simple ratio for vegetation (MSR)** | (R800 - R445)/ (R680 - R445) | Improves the identification of vegetation stress and chlorophyll levels. | [37] |
| **Normalized difference vegetation index (NDVI)** | (R842-R665)/(R842+R665) | Measures vegetation health | [38] |
| **Modified chlorophyll absorption in reflection index (MCARI)/ Optimized soil adjusted vegetation index (OSAVI)** | [(R700 - R670) - 0.2(R700 - R550)] * (R700/ R670) * (1+0.16) * (R800 - R670)/ (R800+R670+0.16) | Identification of vegetation stress and health in diverse environments. | [39] |
| **Beison Datt Vegetation Index (DATT)** | R850/ (R550 - R708) | Distinguishing variations among vegetation types | [37] |

**2.2.2. Endmembers (EMs) selection and spectral data collection.** In this particular study, three spectra relevant to the Robusta coffee CS were identified and utilized in the MESMA experiment. According to Kebede et al. [45], the three EMs (Fig 4) were classified as follows: (1) Robusta coffee with agroforestry, where tall (18 m on average) fruit and timber trees were co-existing with the coffee crop. In this CS, coffee is an understory layer. The primary timber trees in this CS included *Grevillea robusta*, *Albizia adianthifolia*, *Ficus natalensis*, *Maesopsis eminii*, and *Markhamia lutea* whereas the fruit trees were *Artocarpus heterophyllus*, *Mangifera indica*, *Persea americana*, *Carica papaya*, and *Citrus reticulata*, This CS also included some shrubs such as *Hibiscus syriacus*, (2) Robusta coffee with banana. This CS was dominated by Robusta coffee and banana. Additionally, intercrops of short fruits and vegetables were also co-existing as understory layer. The main short fruits and intercrops in this category included *Citrus sinensis*, *Theobroma cacao*, *Manihot esculenta*, *Dioscorea bulbifera*, *Zea mays*, and *Ipomoea batatas*, and (3) Robusta coffee under full sun without shade trees layer. The spectral data for these EMs were collected using *in-situ* hyperspectral measurements that are fully described in [45]. A total of 60 purposefully selected field plots representing Robusta coffee CS were identified within the study area. From these plots, a total of 1,260 leaf spectra were collected using a FieldSpec HandHeld 2™ Spectroradiometer [46]. This non-imaging sensor measures electromagnetic radiation in the 325–1075 nm range with a 25-degree full conical field of view [46]. For each of the three target endmembers, 60 samples were obtained. Leaf-level spectral data were collected between January 1st and February 28th, 2023, from 20 plant species, with three leaf spectra measured per species, as well as five bare soil samples. Prior to measurement, the spectroradiometer was optimized and calibrated using a Spectralon white reference panel (~100% reflectance) to correct for solar illumination and atmospheric variability. Calibration was repeated every 20 readings to ensure consistent and reliable data [47]. To minimize measurement error, the spectroradiometer was held 5–7 cm above the adaxial (upper) leaf surface, depending on leaf size [48]. All readings were taken under clear, sunny conditions between 10:00 a.m. and 2:00 p.m. local time (GMT + 3). Each leaf spectrum was recorded three times to reduce handling error and improve accuracy, resulting in a total of 1,260 high-quality spectral measurements [48]. Subsequently, these samples were filtered using the 'Noise Filtering' function and smoothed with the 'Savitzky–Golay' filter utilizing the *'hsdar'* package [49] in R software [32]. Spectral resampling to match the S2 sensor configuration was performed using the *'spectralResampling* function in the *'hsdar'* package [49] in R software [32]. The resulting mean reflectance values for each band were then used as input EM spectra for the MESMA implemented in R

Robusta with agroforestry (AFS)  Robusta with banana  Robusta under full sun

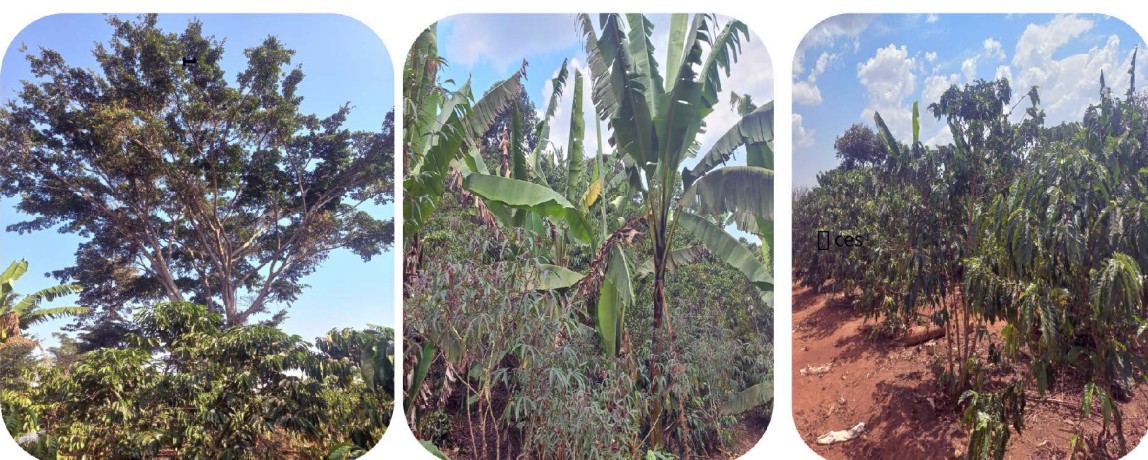

**Fig 4. Field photographs of the three Robusta coffee cropping systems (CS): Robusta coffee with agroforestry, Robusta coffee with banana and Robusta coffee under full sun.**

software [32] via the *'mesma()'* function from the RStoolbox package [50] which enable sub-pixel unmixing and model evaluation.

**2.2.3. Robusta coffee cropping systems (CS) discrimination using multiple endmember spectral mixture analysis (MESMA).** A 3-month median S2 image composite (01/01/2023–28/02/2023), was pre-processed in GEE following the procedure detailed in Section 2.2.1. This image composite was chosen due to its temporal correspondence with the in situ spectral data collection period. A sub-pixel MESMA model, which is a linear mixture of pure spectral EMs, [19] was applied to classify the three Robusta coffee CS. This model allowed the calculation of the fractional abundance of the CS [51]. For each pixel, MESMA iteratively runs multiple candidate models, with the best model determined based on identified pixel fractions, their residuals, and the smallest root mean square error (RMSE) compared to the pixel's spectral curve [21]. These models are then applied to the entire image on a per-pixel basis. In this study, the MESMA algorithm, integrated into the 'RStoolbox' package [52] in R software [32] was used. The 'RStoolbox' package for MESMA employs the non-negative least squares (NNLS) regression through a sequential coordinate-wise algorithm. The MESMA execution was performed using default settings, including the NNLS method, 400 iterations, and a tolerance of $1 \times 10^{-9}$. The MESMA algorithm's outputs consist of individual bands representing estimated per-pixel EM abundance fractions and the probability of occurrence for each tested EM, ranging from 0 to 1. We employed spectral mixture analysis (SMA) to generate the fractional abundance maps for three Robusta coffee CS, following the recommendations of Salih [53]. We also performed a threshold sensitivity to compute the percentage of pixels exceeding fractional abundance thresholds of 0.3, 0.4, and 0.5 for Robusta coffee with agroforestry, Robusta coffee with banana, and Robusta coffee under full sun, respectively. We then assessed the spectral separability among endmembers using the spectral angle mapper (SAM) approach as suggested by Salih [53]. The SAM computes the angle (in degrees) between spectral vectors in n-dimensional space, where smaller angles indicate higher similarity. Although SAM is not a classifier, it is widely used to quantify spectral similarity and assess the distinctiveness of LULC classes [54,55]. Based on established guidelines, spectral angles less than 10–15° indicate high similarity, angles between 15–30° suggest moderate separability, and angles greater than 30° reflect low similarity or high separability between classes [56].

**2.2.4. Classification accuracy assessment.** Accuracy evaluation is crucial in generating thematic maps using remotely sensed data [57]. This process requires high-quality reference datasets sampled at appropriate spatial and temporal scales [58]. To assess the two-step classification approach for creating Robusta coffee CS maps, comparisons were made with ground-based verification point data.

The evaluation of classification outcomes from both stages, including major LULC and Robusta coffee cropland mapping, involved comparing random validation pixels for accuracy. For major LULC classification, 70% of the extracted pixels were used for training and 30% for validation. However, we acknowledge that the validation relied on polygons and pixels from the same period and region, which may have led to an overestimation of accuracy [59], as independent temporal or spatial datasets were not available for this study. In the absence of percentage coverage data for certain Robusta coffee CS classes, confusion matrix methods were used to validate MESMA results. Each pixel in the MESMA map was assigned to the land cover class with the highest fractional coverage, ensuring a robust accuracy assessment. For MESMA classification, only 228 independent ground-based points collected from the three Robusta coffee CS field plots were used for assessing the classification accuracy. Specifically, the validation of the classification involved the use of accuracy assessment metrics derived from confusion matrices. These metrics encompass overall accuracy (OA), user's accuracy (UA), producer's accuracy (PA) and Kappa statistics (K). The PA represents the accuracy of correctly classified samples within each category, while the user's accuracy (UA) denotes the proportion of correctly classified samples for a specific class. The Kappa statistic (K) measures the discrepancy between the actual agreement of a classified map and the agreement that would be expected by chance, based on comparison with reference data [60]. The root mean square (RMSE) evaluates the average magnitude of prediction errors by measuring the square root of the average squared differences between predicted and observed values. Lower RMSE indicates better model accuracy [61]. Additionally, the

overall accuracy (OA) reflects the accuracy of correctly classified samples among all samples of the Robusta coffee CS [23]. Area estimates for each class were adjusted using the reference sample to correct for map classification errors and were accompanied by 95% confidence intervals (CIs) to quantify the uncertainty in the estimated areas. All analyses were performed in R following recommended practices. This approach follows the accuracy assessment and area estimation best practices as recommended by Olofsson et al. [62], which emphasize the importance of integrating sampling design, response design, and consistent analysis for transparent land change validation.

## 3. Results

### 3.1. Major land use and land cover (LULC) mapping

The study area's major LULC classes (five classes) were mapped with an OA of 93.5% (Fig 5 and Table 4). The study area was dominated by mixed agricultural and grassland class, which covered 75.1% of the total area. In contrast, the built-up areas account for only a small fraction (1.7%). Robusta coffee cropland accounts for 8.7% of the total area.

### 3.2. Robusta coffee cropping systems (CS) classification using multiple endmember spectral mixture analysis (MESMA)

Fig 6, illustrates the spectral responses of the three chosen EMs within the eight resampled S2 bands, aligning with the wavelength range of the spectroradiometer instrument employed in this study. Four S2 bands (red-edge2, red-edge3, NIR, and NIRn), noticeable inter-category variations in EMs reflectance values were observed. Furthermore, the correlation between bands Red-edge3, and NIRn had a p-value < 0.0008, indicating a statistically significant correlation between the three endmembers, illustrating spectral distinctiveness. Conversely, the visible bands (blue, green, and red) and red-edge1 did not exhibit any significant differences across the three EMs.

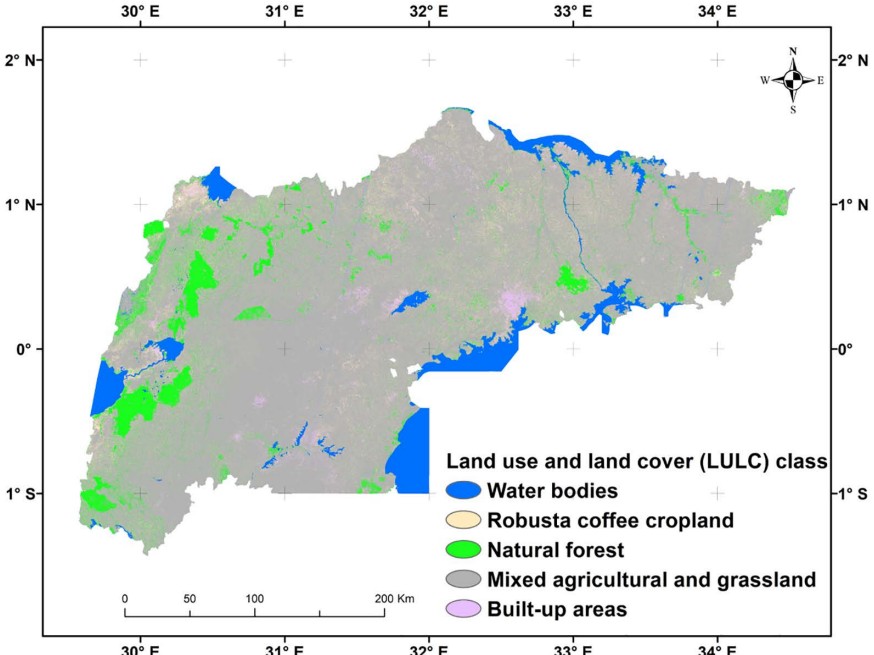

**Fig 5. A graphical representation of the predicted major land use and land cover (LULC) classes of the study area, generated using Sentinel-2 (S2) data and a random forest (RF) classifier in Google Earth Engine (GEE).**

**Table 4. Major land use and land covers (LULC) area coverage, user's accuracy (UA), producer's accuracy (PA), and overall accuracy (OA).**

| Major LULC type | Area coverage (%) | | UA (%) | PA (%) |
|---|---|---|---|---|
| | Area (ha) | % | | |
| Natural forest | 708,886.7 | 7.5 | 96.2 | 95.5 |
| Mixed agricultural and grassland | 7,118,342.6 | 75.1 | 93.0 | 98.6 |
| Built-up areas | 157,128.1 | 1.7 | 80.0 | 94.1 |
| Robusta coffee cropland | 827,924.0 | 8.7 | 95.2 | 59.4 |
| Water bodies | 667,528.9 | 7.0 | 100 | 100 |
| Total | 9,479,810.3 | | OA = 93.5% Kappa = 86% | |

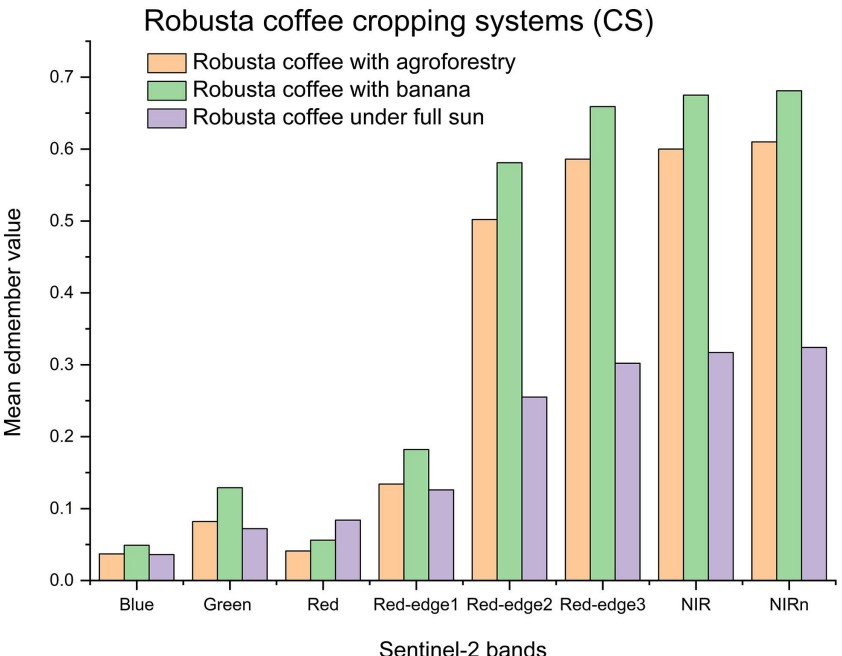

**Fig 6. Three endmembers and their respective values derived from the resampled eight Sentinel-2 (S2) spectral bands used in the multiple endmember spectral mixture analysis (MESMA) for mapping Robusta coffee cropping systems (CS). The eight bands correspond to the blue, green, red, red edge (RE1), RE2, RE3, near infrared (NIR), and** narrow-band near-infrared (**NIRn) waveband of S2.**

Fig 7 depicts the distribution and frequency of pixel fractions for the three EMs used in this study based on the unmixing result on S2. In the study area, Robusta coffee with agroforestry occupied the majority of the pixels, followed by Robusta with banana and Robusta under full sun EMs. Robusta coffee with agroforestry occupied numerous pixels with relatively large fractions, however, fewer pixels contained Robusta coffee with banana or Robusta coffee under full sun with these fractions generally below 0.5. The high fractional abundance of Robusta coffee with agroforestry pixels ranged between 0.1 and 0.6-pixel fractions, with > 65% of pixels containing zero fractions of the two Robusta coffee CS.

The proportion of Robusta coffee under full sun EM was relatively low in comparison to the other two EMs, as evidenced by the few pixels representing the Robusta coffee under full sun fraction. The majority of pixels had a low RMSE (< 0.05), whereas the highest RMSE over all pixels was 0.078. The average RMSE (0.021) indicated a good model fit for

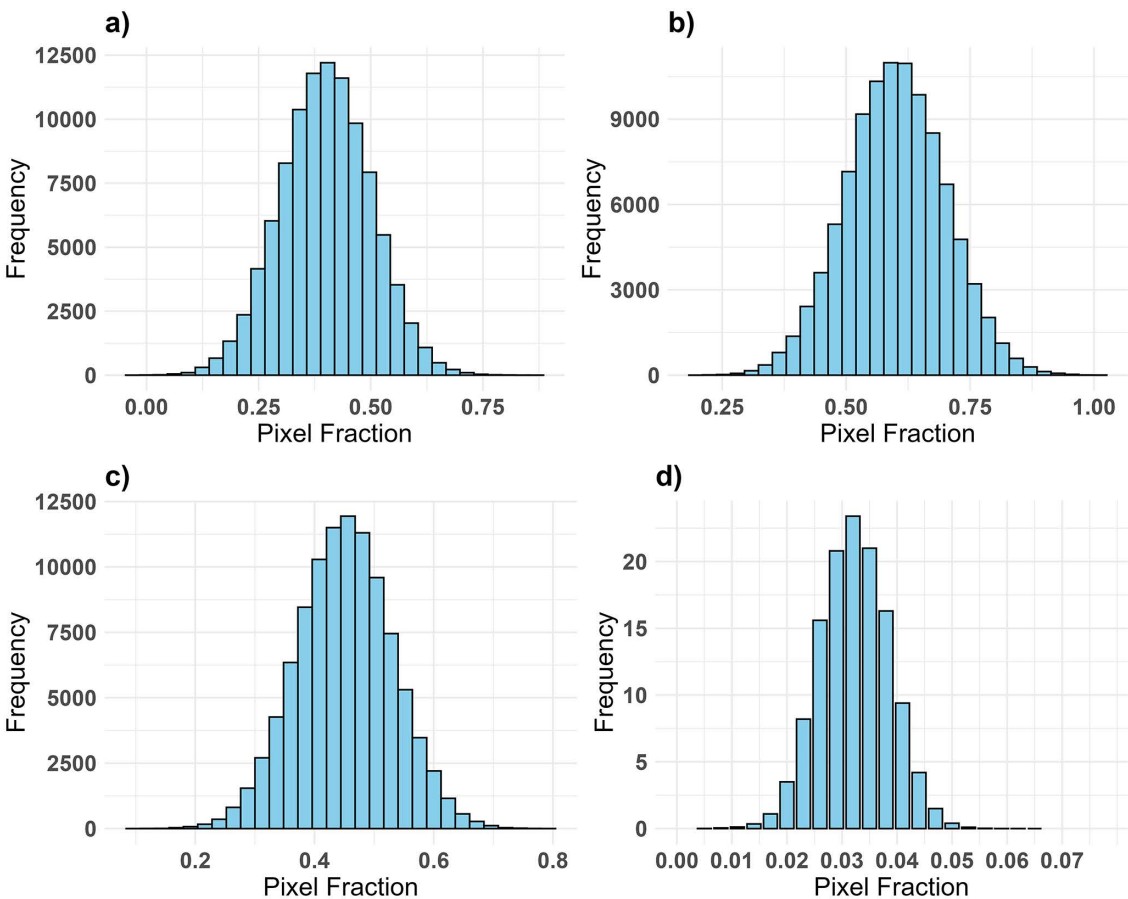

**Fig 7. Frequency of the pixel fractions values within the endmember fraction images of a) Robusta coffee with agroforestry b) Robusta coffee with banana, c) Robusta coffee under full sun, and d) distribution of the root mean square error (RMSE) pixel values.**

MESMA and relatively small EM prediction errors across the entire study area. MESMA created four fraction images for the three EMs, along with their corresponding RMSEs.

Fig 8 depicts the S2-based classification results from the MESMA of the fraction images of the three EMs. The results indicate that the RMSE was typically low throughout the study area, except for the northeast and southwest areas. The fractional abundance distribution of Robusta coffee CS types in the study area varied, with Robusta coffee with banana pixels ranging between 0.5 and 0.6 dominating the southwest and Robusta coffee with agroforestry in the middle and northeastern parts of the major Robusta coffee growing-regions. The threshold sensitivity analysis (Fig 9a) showed that the percentage of pixels classified under each of the three Robusta coffee CS at thresholds of 0.3, 0.4, and 0.5. The Robusta coffee with banana CS maintained the highest pixel purity, followed by Robusta coffee under full sun, while Robusta coffee with agroforestry showed the steepest decline due to spectral mixing. The SAM analysis (Fig 9b) revealed very high spectral separability (50°) between Robusta coffee with agroforestry and banana CS, indicating clear spectral distinction. A moderate separability (30°) was observed between Robusta coffee with banana and Robusta coffee under full sun CS, while Robusta coffee with agroforestry and Robusta coffee under full sun showed low separability (20°), suggesting overlapping spectral characteristics.

The classification results of the three types of Robusta coffee CS presented in Table 5 and Fig 10, indicated 89.9% classification accuracy for all Robusta coffee CS classes and disentangled as follows: Robusta coffee with agroforestry

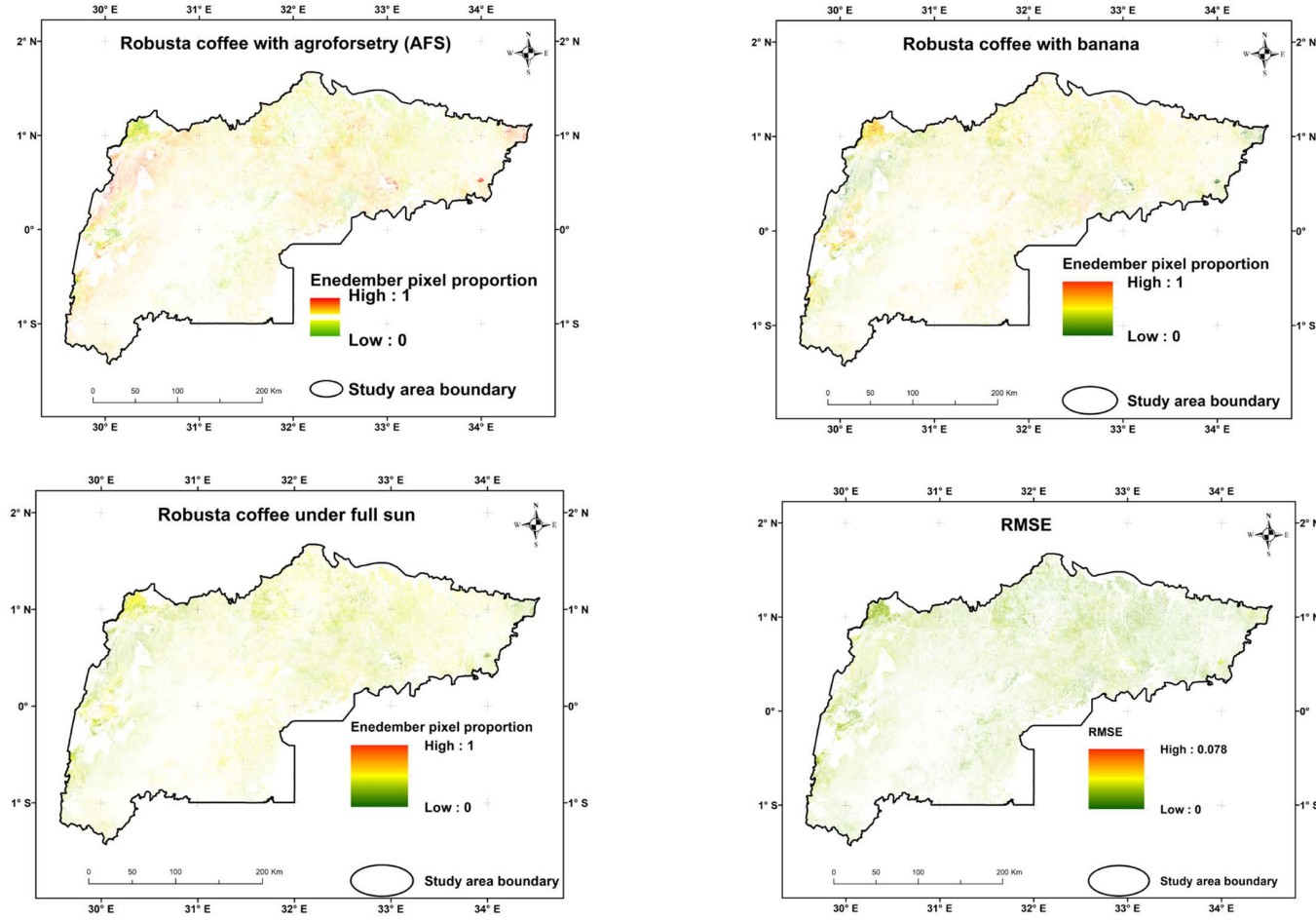

**Fig 8. Illustration of the multiple endmember spectral mixture analysis (MESMA) classification findings, exhibiting the fraction images of the three endmembers from the lowest 0 (green) to the maximum proportion of 1 (red), as well as the root mean square error (RMSE) from the lowest 0 (green) to the highest 0.078.**

covers 63.7% of the total coffee CS area and has a UA of 91.3.7%, Robusta coffee with banana covers 24.8% with a UA of 88.5%, and Robusta coffee under full sun covers 11.5% with a UA of 91.2%. Moreover, the corrected area estimates indicate that Robusta coffee with agroforestry is the most widespread CS, covering 468,377 hectares (95% CI: 431,243–505,511 ha), followed by Robusta with banana at 189,814 hectares (95% CI: 172,879–206,749 ha). Robusta coffee under full sun is the least extensive, with 79,658 hectares (95% CI: 69,394–89,922 ha).

## 4. Discussion

This study employed the RF algorithm, utilizing the high spatiotemporal resolution of S2 imagery integrated within the GEE platform to classify major LULC, including Robusta coffee cropland. To enhance the classification accuracy, five VIs were incorporated together with S2 bands. These indices, selected for their sensitivity to vegetation structure, chlorophyll content, and canopy moisture [63]. These plant morphological and physiological traits could considerably vary among our major LULC classes, and since they were mimicked by the VIs. Studies have shown that when VIs are combined with the spectral bands, the classification accuracy can be reasonably improved [64,65], particularly in such a complex landscape that is characterized by fragmented and small-scale crop fields (<0.5 ha). The use of multi-date and fine spatial resolution

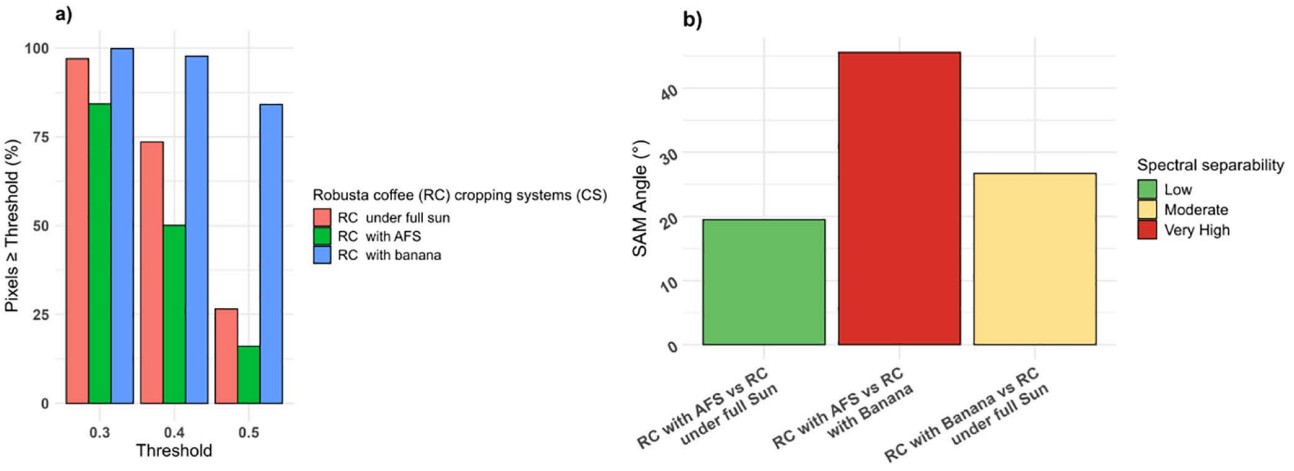

**Fig 9. Proportion of pixels exceeding different threshold values (0.3, 0.4 and 0.5) for each Robusta coffee cropping system (CS), illustrating threshold sensitivity a), and spectral angle mapper (SAM) values between each pair of Robusta coffee CS, quantifying spectral separability b).**

**Table 5. Mapped and corrected area estimates (ha) of Robusta coffee cropping systems (CS), along with PA), UA, OA and CI, are producer's, user's, overall accuracies and confidence intervals, respectively.**

| Robusta coffee cropping systems (CS) | Robusta coffee CS land cover | | UA | PA | Corrected area (ha) | 95% CI (ha) |
|---|---|---|---|---|---|---|
| | Mapped area (ha) | % | | | | |
| **Robusta coffee with agroforestry** | 527,007.2 | 63.7 | 91.3 | 89.0 | 468,377 | [431,243–505,511] |
| **Robusta coffee with banana** | 205,472.2 | 24.8 | 88.5 | 92.4 | 189,814 | [172,879–206,749] |
| **Robusta coffee under full sun** | 95,444.6 | 11.5 | 91.2 | 83.8 | 79,658 | [69,394–89,922] |
| **Total** | 827,924.0 | 100.0 | OA = 89.9%<br>Kappa = 83.0% | | 737,849 | |

S2 imagery enabled the detection of seasonal vegetation dynamics, contributing to highly accurate major LULC classification. However, some classes, such as the Robusta coffee cropland, showed a relatively lower classification accuracy (PA = 59.4%) compared to other land covers. This is likely due to the spectral similarity of Robusta coffee cropland with surrounding vegetation and co-existing species. Also, Robusta coffee cropland is a mixture of shaded and unshaded CS that have different spectral characteristics, which could not have been distinguished at a pixel level.

Many recent studies have reported the successful application of remote sensing techniques for crop classification and mapping [59,66,67], including discriminating Robusta coffee CS [45]. The precision of cropland mapping using S2 imagery has been particularly emphasized. Similarly, Gomez et al. [68] in New Caledonia and Mosomtai et al. [7] in Kenya successfully mapped coffee crop among other associated LULC classes using S2 imagery [69]. Additionally, Escobar-López et al. [13] detected shaded and unshaded coffee crops in Mexico with an OA of 95% using S2 and other auxiliary climatic variables.

The classification of Robusta coffee CS demonstrated high reliability, with an OA of 89.9% and a Kappa coefficient of 83.0%, indicating strong agreement with reference data. UA and PA ranged from 88.5% to 91.3% and 83.8% to 92.4%, respectively. The corrected area estimates revealed that Robusta coffee with agroforestry is the dominant CS, followed by Robusta coffee with banana and Robusta coffee under full sun. These accuracy-adjusted estimates provide a more reliable assessment of the true spatial distribution of each Robusta coffee CS than uncorrected map outputs. Therefore,

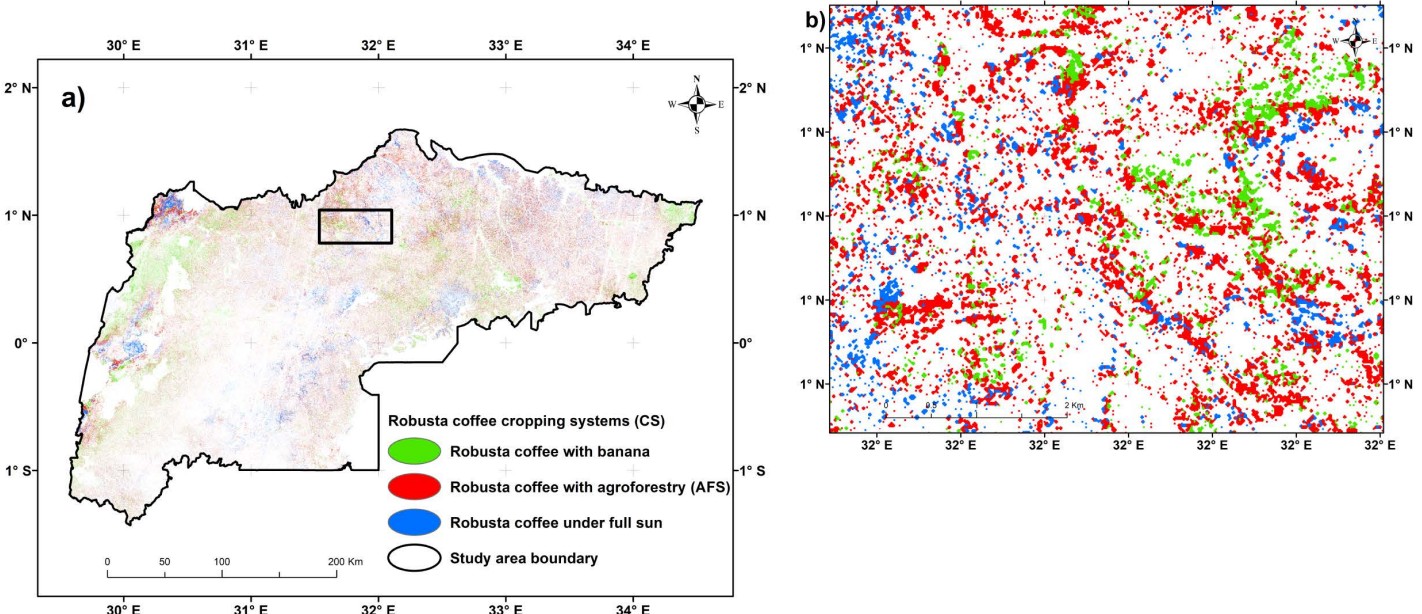

**Fig 10. A graphical representation of the fractions of the three endmembers (EMs) i.e., Robusta coffee with agroforestry, Robusta coffee with banana, and Robusta coffee under full sun that were derived from multiple endmember spectral mixture analysis (MESMA) with a) showing the Robusta coffee cropping system (CS) classification map of the entire study area, b) zoomed section of the study area with the top black box.**

the corrected area estimates offer a more accurate and defensible basis for understanding the landscape-level prevalence of each Robusta coffee CS. The slight variation may likely be attributed to the complexity of the heterogeneous landscape and the diverse crop composition among smallholder farmers. This variability introduces significant spectral and textural differences, making major LULC classification more challenging. Importantly, mapping Robusta coffee CSs extends beyond simple land-cover classification. As noted by Cassamo et al. [70], detailed characterization of cropping systems forms the foundation for assessing ecosystem services and sustainability within smallholder agroecosystems. This highlights the broader policy and environmental relevance of our findings for coffee-producing regions, where spatially explicit insights can inform sustainable land management and climate-resilient agricultural planning. Additionally, the prevalence of mixed Robusta coffee CS on small fields (<0.5 ha) in regions like Uganda have a wide range of field sizes, orientations, and shapes, which might have also contributed to the relatively lower classification accuracy. Notably, the UA (commission error) for Robusta coffee cropland mapping was higher than the PA (omission error), indicating a trade-off between omission and commission errors in the classification process. The relatively higher omission errors in our major LULC classification experiment could also be due to the small field size and several other confounding landscape factors. This approach led to lower commission errors, where other major LULC areas were misclassified as Robusta coffee cropland. This challenge of balancing PA and UA aligns with findings from previous studies [43,71].

The resampled S2 red-edge and near infrared bands (5, 6, 7, and 8a) showed significant variations in the mean reflectance values of Robusta coffee CS EMs. In contrast, the visible bands (1, 2, and 3) and red-edge1 (4) did not display noteworthy differences across the three Robusta coffee CS EMs. This highlights the discernible spectral characteristics within specific bands and underscores the efficacy of these wavelengths in capturing variations among EMs, aligning with previous studies emphasizing the importance of selected S2 bands in species identification [72]. Furthermore, the observed differences among Robusta coffee CS EMs were attributed to the distinct spectral signatures of chlorophyll,

nitrogen, phosphorus, and potassium in Robusta coffee and its co-existing plants for each EM group, which exhibit distinguishable characteristics in the red-edge region [73].

The results of the MESMA showed that most pixels exhibited low RMSE values, typically below 0.05, with the highest RMSE value of 0.078, encompassing nearly the entire study area. The mean RMSE of 0.021 suggests that MESMA provides a good model fit with minimal prediction errors across the study area; however, we acknowledge that a low RMSE does not necessarily equate to high classification accuracy. Moreover, MESMA accuracy assessments, using independent validation data, demonstrated promising results, with low omission errors. Although alternative sub-pixel classification approaches, such as support vector machine (SVM) [74] and deep learning [75], could have also been applied in this study; but a direct comparison between MESMA and these methods in mapping coffee CS was beyond this study scope. Future studies should explore the performance of different sub-pixel classifiers in mapping Robusta coffee CSs.

The two-step approach, combining RF for major LULC mapping and MESMA for mapping Robusta coffee CS, improved accuracy by minimizing spectral confusion and suppressing spatial constraints from other classes; however, this improvement was not empirically tested within this study. Like our approach, spectral sub-pixel approaches, commonly employed in mapping landscape dynamics involving moderate- and low-resolution imagery within intricate and diverse landscapes, aim to overcome the limitations associated with spatial resolution. These approaches aim to estimate the relative proportions of different constituents within a pixel. They consider a single pixel as a representation and identify pure sub-pixel classes known as "endmembers". Schmitt-Harsh et al. [76] utilized spectral mixture analysis with aerial photographs, which resulted in varying accuracies when mapping coffee agroforestry systems. Similarly, Bispo et al. [77] employed spectral linear mixing but encountered limitations regarding plant heterogeneity, varying plant sizes, diverse management systems, and inconsistent spacing, achieving an OA of 67%. The strength of our approach is that we used a 2-step approach that aims to provide a detailed fractional coverage of three Robusta coffee CS. However, our study area represents a highly heterogeneous cropping system, where the structural and compositional diversity of Robusta coffee CSs influences both classification performance and ecological outcomes. This relationship aligns with broader agroecological evidence showing that variations in cropping system heterogeneity strongly affect remote sensing detectability as well as landscape-scale ecological functions [5,13]. However, while spectral sub-pixel approaches offer an improved capability to capture landscape heterogeneity compared to pixel-based methods, they still face constraints related to cloud cover and tend to be more time-intensive.

The fractional abundance distribution of Robusta coffee CS classes in the study area varied, with Robusta coffee with banana pixels ranging between 0.5 and 0.6 dominating the southwest and Robusta coffee with agroforestry pixels ranging between 0.5 and 1.0-pixel fractions, with > 65% of pixels in the middle and northeast.

In evaluating the classification robustness, the threshold sensitivity analysis revealed that Robusta coffee with banana consistently retained the highest pixel purity across all thresholds, followed by Robusta coffee under full sun, while Robusta coffee with agroforestry exhibited the steepest decline. This suggests that Robusta coffee with banana has more spectrally distinct and homogeneous coverage, whereas Robusta coffee with agroforestry is more spectrally mixed due to vegetation complexity. As the fractional threshold increased from 0.3 to 0.5, the decline in pixel percentages, particularly for Robusta coffee with agroforestry and Robusta coffee under full sun, indicates growing uncertainty in abundance estimates at stricter thresholds. Supporting these findings, the SAM analysis showed very high spectral separability (50°) between Robusta coffee with agroforestry and Robusta coffee with banana, indicating a considerable distinction. Moderate separability (30°) was observed between Robusta coffee with banana and Robusta coffee under full sun, while Robusta coffee with agroforestry and Robusta coffee under full sun displayed low separability (20°), suggesting greater spectral similarity. These results underscore the ability to discriminate Robusta coffee with agroforestry from Robusta coffee with banana, while distinguishing Robusta coffee with agroforestry from Robusta coffee under full sun remains more challenging due to spectral overlap.

However, caution is advised in interpreting these findings due to challenges in determining optimal endmembers that represent the heterogeneity of Robusta coffee CS. Spectral variations within the same class may be influenced by factors such as tree ages, abiotic stresses, and insect pests such as black coffee twig borer (BCTB) that damages the coffee leaves. Also, it is possible that some of the spectral variations in our EMs were obscured by the area sampled (scanned) by the spectroradiometer. This frequently results in uncertainty, especially when taking into account the temporal variance of the EM spectra recorded at various locations and times. Furthermore, socio-economic variables, such as farmer practices, crop management intensity, and yield differences, could also contribute to unexplained spectral variability in the EMs. While such datasets were currently not spatially available, integrating these information in future work could provide a more comprehensive understanding of Robusta coffee CS spectral heterogeneity for effective and accurate classification and mapping.

Specifically, integration of different Robusta coffee CS as an endmember contributed to accurate Robusta coffee CS discrimination. However, caution is warranted when applying this approach in different spatial or temporal settings due to potential variations in brightness effects. Also, this study was based on Robusta coffee CS EM data collected only during a dry season (January to February). Seasonal variations in vegetation phenology, pest damage, and other biotic or abiotic stressors can significantly alter the spectral characteristics of both the major LULC classes and the three Robusta coffee CS [73]. These changes reduce their spectral separability and hinder accurate mapping when relying on a single image or spectral sampling from only one season. Therefore, future studies should incorporate multi-seasonal imagery and EMs derived from both dry and wet seasons, as well as from varying stress conditions, to enhance the robustness and reliability of classification accuracy. Similarly, validation polygon samples were obtained from the same period and region to assess the performance of our classification approach. This approach, while practical, may lead to a slight overestimation of classification accuracy [59]. Future studies should employ independent validation datasets collected across different time periods and geographic locations to enhance the reliability and robustness of the classification accuracy assessment. While this study primarily relied on S2 imagery for mapping Robusta coffee CS, it is important to acknowledge the role of in situ spectral data. Although not explicitly compared here, in situ data provide high-resolution spectral detail at the plot level but are limited in spatial coverage and temporal frequency. In contrast, S2 offers consistent and scalable imagery, making it more suitable for operational mapping across large regions.

A key limitation of this study lies in the class imbalance within the training dataset, particularly concerning land cover types such as water bodies and built-up areas. These classes occupy relatively small spatial extents in the study area, leading to limited sample sizes. This imbalance may have adversely affected classification accuracy and constrained the model's ability to generalize across diverse agroclimatic zones. Although the tree Robusta coffee CS were distinctly identified and classified using in situ hyperspectral data in Kebede et al. [45], spectral characteristics of other tree crops, which could be confused with Robusta coffee, and the possible presence of Arabic coffee could have compromised the purity of the ENM applied in this study. Another shortcoming of this study is the inclusion of agriculture and grassland in one major LULC class. This could have caused spectral confusion in this class as the two land cover types, i.e., agriculture and grassland, have distinct spectral characteristics. As long as the spatial resolution of the Google Earth background image (30 m) is concerned, and based on our field observations, a clear separation between agriculture and grasslands was not possible in our study area. Although representative pixels were carefully selected using field data and medium-resolution imagery, we recommend that future research should mitigate these limitations by increasing sample sizes for underrepresented classes and using higher-resolution imagery.

## 5. Conclusions

Our 2-step classification approach improves the accuracy of discriminating and mapping Robusta coffee CS in the main Robusta coffee growing-regions of Uganda. The study demonstrated the effectiveness of S2 composite image, the RF classifier, and MESMA in discriminating among the three Robusta coffee CS classes. Importantly, producing a Robusta coffee cropland mask before employing MESMA enabled a focused and spatially adjusted spectral unmixing process.

Additionally, extracting EMs from in-situ spectrometric data provided a more realistic spectral array for the three Robusta coffee CS classes. It is recommended that future research should compare the effectiveness of using Robusta coffee CS EMs sourced from various image datasets employing different EM selection techniques, such as endmember average root mean squared error (EAR), minimum average spectral angle (MASA), and count-based endmember selection (CoB). Moreover, we were able to assess Robusta coffee CS fractional coverage at a sub-pixel scale for three different Robusta coffee CS. Future research should also evaluate the comparative performance of the above-mentioned EMs derived from different datasets and selection strategies in classifying different coffee CS. Furthermore, exploring seasonal or multi-temporal composites, incorporating hyperspectral imagery, or integrating drone-based observations could open new avenues for refining Robusta coffee CS maps that are important for informed decision making regarding coffee pest management, harvest monitoring, implementing sustainable and effective agronomic practices, and ecosystem services.

## Acknowledgments

The authors sincerely appreciate the support and assistance provided by the Uganda National Coffee Research Institute (NaCORI) and the National Forestry Resources Research Institute (NaFORRI) for facilitating field data collection.

## Author contributions

**Conceptualization:** Getachew Kebede, Bester Tawona Mudereri, Henri E. Z. Tonnang, Elfatih M. Abdel-Rahman.

**Data curation:** Getachew Kebede.

**Formal analysis:** Getachew Kebede.

**Funding acquisition:** Henri E. Z. Tonnang, Elfatih M. Abdel-Rahman.

**Investigation:** Getachew Kebede, Bester Tawona Mudereri, Onisimo Mutanga, Tobias Landmann, John Odindi, Natacha Motisi, Fabrice Pinard, Henri E. Z. Tonnang, Elfatih M. Abdel-Rahman.

**Methodology:** Getachew Kebede, Bester Tawona Mudereri, Onisimo Mutanga, Tobias Landmann, John Odindi, Natacha Motisi, Fabrice Pinard, Henri E. Z. Tonnang, Elfatih M. Abdel-Rahman.

**Project administration:** Fabrice Pinard, Henri E. Z. Tonnang, Elfatih M. Abdel-Rahman.

**Resources:** Fabrice Pinard, Henri E. Z. Tonnang, Elfatih M. Abdel-Rahman.

**Software:** Getachew Kebede.

**Supervision:** Bester Tawona Mudereri, Onisimo Mutanga, Tobias Landmann, John Odindi, Natacha Motisi, Henri E. Z. Tonnang.

**Validation:** Getachew Kebede, Bester Tawona Mudereri, Onisimo Mutanga, Tobias Landmann, John Odindi, Natacha Motisi, Fabrice Pinard, Henri E. Z. Tonnang, Elfatih M. Abdel-Rahman.

**Visualization:** Getachew Kebede.

**Writing – original draft:** Getachew Kebede.

**Writing – review & editing:** Getachew Kebede, Bester Tawona Mudereri, Onisimo Mutanga, Tobias Landmann, John Odindi, Natacha Motisi, Fabrice Pinard, Henri E. Z. Tonnang, Elfatih M. Abdel-Rahman.

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
