## [Decision Letter · Decision Letter 0]

29 May 2025

Dear Dr. Aragaw,

Thank you for submitting your manuscript to PLOS ONE. After careful consideration, we feel that it has merit but does not fully meet PLOS ONE’s publication criteria as it currently stands. Therefore, we invite you to submit a revised version of the manuscript that addresses the points raised during the review process.

We look forward to receiving your revised manuscript.

Kind regards,

Tzen-Yuh Chiang

Academic Editor

PLOS ONE

4. Please note that PLOS ONE has specific guidelines on code sharing for submissions in which author-generated code underpins the findings in the manuscript. In these cases, we expect all author-generated code to be made available without restrictions upon publication of the work. Please review our guidelines at https://journals.plos.org/plosone/s/materials-and-software-sharing#loc-sharing-code and ensure that your code is shared in a way that follows best practice and facilitates reproducibility and reuse.

5. Thank you for stating in your Funding Statement:

 [Getachew Kebede was supported by the In-Region Postgraduate Scholarship from the German Academic Exchange Service (DAAD). The authors sincerely appreciate the financial support provided by the European Union (EU) for the project “Robusta coffee agroforestry to adapt and mitigate climate change in Uganda” (GCCA+-Global Climate Change Alliance & DESIRA, Project/Grant No. FOOD/2021/427-759). Additional funding was received from the Swedish International Development Cooperation Agency (Sida), the Swiss Agency for Development and Cooperation (SDC), the Australian Centre for International Agricultural Research (ACIAR), the Norwegian Agency for Development Cooperation (Norad), the Federal Democratic Republic of Ethiopia, and the Government of the Republic of Kenya. The project also benefited from funding by the Centre de Coopération Internationale en Recherche Agronomique pour le Développement (CIRAD).].

6. Thank you for uploading your study's underlying data set. Unfortunately, the repository you have noted in your Data Availability statement does not qualify as an acceptable data repository according to PLOS's standards.

7. When completing the data availability statement of the submission form, you indicated that you will make your data available on acceptance. We strongly recommend all authors decide on a data sharing plan before acceptance, as the process can be lengthy and hold up publication timelines. Please note that, though access restrictions are acceptable now, your entire data will need to be made freely accessible if your manuscript is accepted for publication. This policy applies to all data except where public deposition would breach compliance with the protocol approved by your research ethics board. If you are unable to adhere to our open data policy, please kindly revise your statement to explain your reasoning and we will seek the editor's input on an exemption. Please be assured that, once you have provided your new statement, the assessment of your exemption will not hold up the peer review process.

8. We note that Figures 1, 3, 6 and 7 in your submission contain [map/satellite] images which may be copyrighted. All PLOS content is published under the Creative Commons Attribution License (CC BY 4.0), which means that the manuscript, images, and Supporting Information files will be freely available online, and any third party is permitted to access, download, copy, distribute, and use these materials in any way, even commercially, with proper attribution. For these reasons, we cannot publish previously copyrighted maps or satellite images created using proprietary data, such as Google software (Google Maps, Street View, and Earth). For more information, see our copyright guidelines: http://journals.plos.org/plosone/s/licenses-and-copyright.

1. You may seek permission from the original copyright holder of Figures 1, 3, 6 and 7 to publish the content specifically under the CC BY 4.0 license. 

9. Please remove your figures from within your manuscript file, leaving only the individual TIFF/EPS image files, uploaded separately. These will be automatically included in the reviewers’ PDF.

Additional Editor Comments (if provided):

Reviewers' comments:

Reviewer's Responses to Questions

**Comments to the Author**

1. Is the manuscript technically sound, and do the data support the conclusions?

Reviewer #1: No

Reviewer #2: Partly

2. Has the statistical analysis been performed appropriately and rigorously?

Reviewer #1: No

Reviewer #2: Yes

3. Have the authors made all data underlying the findings in their manuscript fully available?

Reviewer #1: No

Reviewer #2: No

4. Is the manuscript presented in an intelligible fashion and written in standard English?

Reviewer #1: No

Reviewer #2: Yes

Reviewer #1: I am of the view that this whole manuscript needs to be rewritten and even re-analyse. i have attached my comment to improve the manuscript. Please let the authors carefully check my comments to improve the manuscripts

Reviewer #2: Overall Comments:

The authors present an interesting and valuable contribution by developing a robusta coffee map for the main growing regions of Uganda. The study employs a two-step approach that combines Random Forest classification with endmember spectral mixture analysis (MESMA), utilizing spectral information collected in the field.

To enhance reproducibility and facilitate comparison with future studies, I recommend that the authors provide a more detailed description of their methodology and improve the validation protocol of their map. Taking into account mapped areas and giving confidence intervals for their figures.

Additionally, it would be beneficial if the authors elaborate on the potential applications of this map.

Finally, in accordance with the journal’s data availability policy, the authors should ensure that all relevant data, namely, the map, calibration/training data, and validation data are made publicly available.

Recommendation:

As per the journal’s policy:

“Authors are required to make all data underlying the findings described fully available, without restriction, and from the time of publication.”

Currently, the training and validation datasets, as well as the final map, do not appear to be accessible. Additionally, the spectral data shared online are not self-explanatory or documented making their use difficult.

Section: Abstract:

Line 27: The accuracy figure cited as 87% does not correspond to any value in the main text. The authors likely meant 89.9%.

Section: Introduction:

The introduction presents the situation of coffee in Uganda and the mapping methodologies, but do not make clear connection between the two. Additionally, it would be good to expand on how the map can be used.

Line 45: The phrase "lagging behind" is vague. Consider specifying which aspects (e.g., productivity, technology adoption, mapping efforts).

Line 65: The resolution seems inverted between Sentinel and Landsat

Section: Data Collection

This section should be clarified. The sampling protocol is not clear. What is the difference between the ROI and representative pixel? The clustering methodologies should be clearly explained and reproducible. Which climate indicators were used? Which source of data was used for the slope, elevation, rainfall, ... The clustering result should be presented on a map.

What is the google earth platform (Goole earth engine, Google earth)? What is the imagery source with a resolution of 30m. Do the authors mean 30cm?

The number of samples used for validation appears imbalanced and insufficient for some classes. For example, only ~20 pixels are used for water bodies and ~16 for built-up areas. These small sample sizes are not statistically robust for accuracy assessment.

Further, more information is needed on the size of the digitalized polygons. There may be spatial correlation between training and validation samples if they are geographically close, which could artificially inflate accuracy metrics. A figure showing digitalized area could help the reader understand the method.

Section: Methods

The categories choice should be discussed. It should be clarified in which categories other tree crops which could be confused with coffee are grouped. The possible presence of arabica coffee should also be addressed.

Line 142: Is the composite made of 2 images or all the January and February images?

Line 157: Pre-processing steps need to be described in more detail to ensure reproducibility (e.g., cloud masking, atmospheric correction, resampling procedures).

Section: Endmember Selection and Spectral Data Collection

A brief description of the spectral data collection process should be provided. The text should clearly state that data were collected during the dry season.

The author should address how/why those systems were selected. Some comment on the fact that the endmembers are not “pure” as they will all have some of the spectra of robusta plantation. The meaning of decomposing a pixel in Robusta full sun and "banana + robusta” should be explained as robusta appear in the two classes. Naively I would expect that each pixel is decomposed into robusta, banana and other types of agroforestry so there is a clear sum to 1. i.e 60% robustra+40% banana. Finally, the “robusta+agroforestry” category seems ill defined had it is a mix of different systems with different spectra. The authors should specify which agroforestry systems are included in this category.

Section: Robusta coffee cropping systems (CS) discrimination using multiple endmember spectral mixture analysis (MESMA)

The purpose of the first paragraph in this section is unclear. Is it used to clean the ground data? How does it link to the process in R.

Please specify the QGIS plugin or function used

Section: Accuracy Assessment

The methodologies should be clearly stated. It is unclear which points were used for validation. While the text states that only ground points were used, based on earlier paragraph it seems a mix of ground points and points from high-resolution imagery interpretation were used. It also appears the validation sample could be taken in the same polygon as the calibration samples. This high correlation usually led to overestimation of the different accuracy metrics.

The current validation seems to be based on sample count. I strongly recommend doing a “map” validation. Where the user and producer accuracy are representative on the map content. Additionally, the validation method should allow the result to come with their corresponding confidence interval. One recommended method is described here:

P. Olofsson and al.: Good practices for estimating area and assessing accuracy of land change.

The validation sample size for coffee seems to be 0.3*340=102. But the count for each coffee type is not specified and potentially small (less than 30).

The author should discuss the fact the coffee has a lower producer accuracy than the other cover. It should be discussed if some specific coffee system is not well represented on the map. As this would impact the quality of the final product combining the 2 maps.

Line 240: The definition is not clear as the difference between class and categories is not explained.

Line 244: RMSE not explained. And OA seems to be defined only for the robusta analysis only.

Section: Results

Line 275: The sentence is difficult to follow and should be rephrased for clarity.

Section: Discussion

Line 329: The citation is repeated

Line 325: As the composite is only based on median of one year of imagery, the statement that seasonal and interannual dynamic is captured should be justified.

Line 330-333: specify the country mapped in other studies

Line 334: without confidence interval and with potentially different validation method the difference between the studies is probably not statistically significant.

Line 340-346: in table 3 UA=95% > PA=59.4% it seems either the texts or table have inverted UA and PA.

Line 358: A low RMSE indicates good model fit but does not necessarily imply high classification accuracy.

Line 362: While the two-step approach may intuitively improve accuracy, this was not empirically tested in the study.

It would be interesting to add a comment on the accuracy once the 2 steps are combined. As it seems the 2 tables are addressing one step at time.

Section: Conclusion

Line 400: The superiority of the proposed approach over a direct classification was not tested and should not be claimed without supporting evidence.

The value of in-situ spectral data extraction versus extraction from Sentinel data was not explicitly assessed in the study. This point should be revised or supported with additional analysis.

Tables and Figures

Table 1:

Consider adding example images for each cropping system.

The definition of "forest" should be refined. Terms like "normal natural forest" are vague. Specify whether this includes primary, secondary, or degraded forest.

Clarify which categories contain non-coffee tree crops.

Overall, class definitions should be made more precise and operational.

Table 2:

Clarify what the numbers in the formula represent. Presumably, wavelength of the different bands?

Figure 5:

Use uniform scale for figures 1-3 and same figure size. Figure d) has a wrong horizontal axis.

Figure 6:

The figure text is too small and hard to read. White on yellow contrast if hard to see. Use the same size for all the figures.

Figure 6:

Some zoom would be interesting.

**Do you want your identity to be public for this peer review?** For information about this choice, including consent withdrawal, please see our Privacy Policy

Reviewer #1: No

Reviewer #2: **Yes: ** Thibaud Vantalon

---

## [Author Response · Author response to Decision Letter 1]

12 Aug 2025

We sincerely thank the Editor and Reviewers for their thorough evaluation and constructive feedback on our manuscript. We have carefully addressed all the comments provided in the decision letter. A detailed, point-by-point response to each comment is included in the attached response document, indicating how and where revisions were made in the manuscript.

We trust that the revisions have improved the quality and clarity of the manuscript. Please do not hesitate to contact us if further clarification is needed.

---

## [Decision Letter · Decision Letter 1]

22 Sep 2025

Dear Dr.  Aragaw,

Thank you for submitting your manuscript to PLOS ONE. After careful consideration, we feel that it has merit but does not fully meet PLOS ONE’s publication criteria as it currently stands. Therefore, we invite you to submit a revised version of the manuscript that addresses the points raised during the review process.

We look forward to receiving your revised manuscript.

Kind regards,

Tzen-Yuh Chiang

Academic Editor

PLOS ONE

Journal Requirements:

Reviewer's Responses to Questions

**Comments to the Author**

Reviewer #3: (No Response)

Reviewer #4: All comments have been addressed

2. Is the manuscript technically sound, and do the data support the conclusions?

Reviewer #3: Yes

Reviewer #4: Yes

3. Has the statistical analysis been performed appropriately and rigorously?

Reviewer #3: Yes

Reviewer #4: Yes

4. Have the authors made all data underlying the findings in their manuscript fully available?

Reviewer #3: Yes

Reviewer #4: Yes

5. Is the manuscript presented in an intelligible fashion and written in standard English?

Reviewer #3: Yes

Reviewer #4: Yes

Reviewer #3: The MS entitled ‘Mapping Robusta coffee (Coffea canephora) cropping systems in Uganda: A two-step pixel and sub-pixel based approach with Sentinel-2 data’ could be important because it advances methods to accurately map Uganda’s heterogeneous Robusta coffee systems, providing critical insights for climate adaptation, policy planning, and improving smallholder farmer livelihoods. The authors aimed to evaluate the use of MESMA with Sentinel-2 imagery to classify major LULC types, isolate Robusta coffee cropland, and map three distinct Robusta coffee cropping systems at a subpixel scale in Uganda. Although most sections of the manuscript are well-developed, certain revisions are necessary before it is ready for publication.

Comments:

1. In line 22, the scientific name of Robusta coffee should be provided immediately after the term 'Robusta coffee,' rather than being repeated later in line 25.

2. The sectional titles lack clear organization: some are included in figure captions, while others are associated with subtitles.

3. In lines 100-101, ‘MESMA to the coffee class only to map the three CS at a subpixel scale’ is confusing to me. It would be nice if the authors would rewrite and clarify it.

4. The socio-economic variables, like farmer practices, yield differences, or management intensity, could affect the results. The discussion would be more solid if the authors addressed these missing variables.

5. The study relies on a static image composite from a single dry season, making it unlikely to capture seasonal vegetation dynamics. The authors should consider revising these discussions or adding additional data to assess seasonal changes in vegetation.

6. Although the authors mention Robusta coffee productivity in the abstract and introduction, the study presents no results addressing it. The authors should either remove this content or provide additional data and analyses to demonstrate the study’s relevance to Robusta coffee productivity.

Reviewer #4: Overall Assessment

This manuscript presents an innovative two-step remote sensing approach for mapping Robusta coffee cropping systems (CS) in Uganda, integrating Random Forest classification of LULC with subpixel MESMA spectral unmixing. The combination of Sentinel-2 imagery with in-situ hyperspectral endmembers is novel for East African coffee systems and provides valuable applied insights for coffee management and climate adaptation.

The work is generally well designed and clearly presented. However, several methodological and interpretive issues require clarification, particularly regarding endmember representativeness, seasonal bias in spectral data, and the absence of comparisons with alternative classifiers or auxiliary predictors (e.g., climatic or topographic variables). Addressing these points will considerably strengthen the manuscript.

Overall, this is a promising and publishable study, provided the authors undertake major revisions.

Strengths

• High relevance: Uganda is a key Robusta producer, and mapping coffee systems has direct implications for adaptation to climate change and value-chain management (Bunn et al., 2019) [https://cgspace.cgiar.org/bitstreams/bd1b904e-271c-4e50-ad35-555321bc02f5/download?utm_source=chatgpt.com]; Bukomeko et al., 2019) [https://doi.org/10.1007/s10457-017-0172-8].

• Novel methodology: The two-step design—Random Forest for LULC followed by MESMA at subpixel scale—demonstrates a creative approach rarely applied to coffee agroforestry mapping.

• Integration of field hyperspectral data: Using in-situ spectra to derive endmembers increases ecological realism and reduces reliance on generalized spectral libraries (Somers et al., 2011)[https://doi.org/10.1016/j.rse.2011.03.003].

• Strong classification results: The reported accuracies (93.5% LULC; 89.9% coffee CS) are notable given the highly heterogeneous smallholder landscape.

• Transparency and data access: The provision of open data resources supports reproducibility and potential reuse by other researchers.

Major Concerns

• Endmember representativeness

o The analysis is based on three endmembers sampled during a single dry season. This may not fully capture the spectral variability introduced by phenology, pest damage, or stress conditions (Mutanga & Skidmore, 2004) [https://doi.org/10.1080/01431160310001654923].

o The authors should discuss how seasonal changes might bias classification and outline future improvements (e.g., inclusion of wet-season spectra).

• Comparison with alternative methods

o MESMA is well established, but no comparison is made with alternative classifiers or subpixel approaches (e.g., SMA, SVM, or deep learning; Kussul et al., 2017) [https://doi.org/10.1109/LGRS.2017.2681128].

o Even if not implemented, the rationale for selecting MESMA over newer approaches should be expanded.

• Validation strategy

o The validation relies on polygons from the same period and region. This may overestimate accuracy (Tariq et al., 2023) [https://doi.org/10.1080/10095020.2022.2100287]. Consideration of independent temporal or spatial datasets would strengthen reliability.

• Contextualization within broader agroecological debates

o The discussion could connect more explicitly with recent literature showing how heterogeneity in cropping systems influences remote sensing detectability and ecosystem outcomes (e.g., Escobar-López et al., 2022 [https://doi.org/10.3390/rs14163847]; Hunt et al., 2020) [https://doi.org/10.3390/rs12122041].

o In particular, Cassamo et al. (2022)[https://doi.org/10.1016/j.agee.2022.108341] emphasize that mapping cropping systems provides a foundation for assessing ecosystem services and sustainability in smallholder agroecosystems. This perspective complements the current work and underscores its potential applications beyond classification, for example in policy design and environmental assessments.

Minor Concerns

• Methodological detail: Some processing steps (e.g., spectral resampling in hsdar, MESMA implementation in RStoolbox) could be described more explicitly for reproducibility.

• Figures: Several figures (Figs. 5–7) would benefit from clearer legends and larger font size.

• Terminology consistency: Ensure consistent use of “coffee cropping systems (CS)” rather than alternating with “coffee systems.”

• References: Update with recent contributions on coffee/agroforestry mapping (Escobar-López et al., 2022; Kelley et al., 2018 [https://doi.org/10.3390/rs10060952]).

• Language polishing: Minor English corrections needed (e.g., “the Robusta CS class was masked” → “the Robusta coffee class was masked”).

The manuscript makes a strong methodological and applied contribution, but requires clarification of uncertainties (seasonality, validation limits), expanded discussion of alternatives, and broader contextualization with recent agroecological literature.

Importantly, the recommendation of “major changes” reflects the need for expanded analysis and framing, not a flaw in the current methodology. With revisions, the study can significantly advance understanding of coffee cropping system mapping in Africa.

**Do you want your identity to be public for this peer review?** For information about this choice, including consent withdrawal, please see our Privacy Policy

Reviewer #3: No

Reviewer #4: No

---

## [Author Response · Author response to Decision Letter 2]

6 Nov 2025

Authors’ Point-by-Point Response to Reviewers’ Comments

Review#3 Comments to the Author

Reviewer #3: The MS entitled ‘Mapping Robusta coffee (Coffea canephora) cropping systems in Uganda: A two-step pixel and sub-pixel based approach with Sentinel-2 data’ could be important because it advances methods to accurately map Uganda’s heterogeneous Robusta coffee systems, providing critical insights for climate adaptation, policy planning, and improving smallholder farmer livelihoods. The authors aimed to evaluate the use of MESMA with Sentinel-2 imagery to classify major LULC types, isolate Robusta coffee cropland, and map three distinct Robusta coffee cropping systems at a subpixel scale in Uganda. Although most sections of the manuscript are well-developed, certain revisions are necessary before it is ready for publication.

Authors’ response

We sincerely thank the reviewer for the positive and constructive feedback. We have carefully addressed all the comments, which significantly improved the quality of our manuscript.

Reviewer #3 – Detailed point-by-point response

Comment 1:

1. In line 22, the scientific name of Robusta coffee should be provided immediately after the term 'Robusta coffee,' rather than being repeated later in line 25.

Authors’ response

Addressed (lines 22 and 25), thank you for the suggestion.

Comment 2:

2. The sectional titles lack clear organization: some are included in figure captions, while others are associated with subtitles.

Authors’ response

Thank you for pointing this out. We have revised the manuscript to ensure consistent and clear organization of sectional titles, separating them from figure captions and aligning subtitles with the main text structure.

Comment 3:

3. In lines 100-101, ‘MESMA to the coffee class only to map the three CS at a subpixel scale’ is confusing to me. It would be nice if the authors would rewrite and clarify it.

Authors’ response

Thank you for the suggestion. We clarified the sentence in lines 106–110 to explain that MESMA was applied only within the Robusta coffee cropland to distinguish the three coffee cropping systems at a sub-pixel scale. The sentence now reads “We first classified the study landscape into five major LULC classes to distinguish Robusta coffee cropland from other land types. After isolating the coffee cropland, we applied MESMA exclusively within the coffee cropland class to further discriminate and map the three coffee CS at a subpixel scale”

Comment 4:

The socio-economic variables, like farmer practices, yield differences, or management intensity, could affect the results. The discussion would be more solid if the authors addressed these missing variables.

Authors’ response

Yes, that is right we agree with the reviewer and included this shortcoming in the Discussion section (lines 583–589) to read “Furthermore, socio-economic variables, such as farmer practices, crop management intensity, and yield differences, may could also contribute to unexplained spectral variability in the EMs. While such datasets were currently not available, integrating these information in future work could provide a more comprehensive understanding of Robusta coffee CS spectral heterogeneity for effective and accurate classification and mapping”.

Comment 5:

The study relies on a static image composite from a single dry season, making it unlikely to capture seasonal vegetation dynamics. The authors should consider revising these discussions or adding additional data to assess seasonal changes in vegetation.

Authors’ response

Thank you for this constructive criticism, which we agreed with. The same concern was also raised by reviewer #4. We discussed this as one of the study limitations in lines 594–596 “Seasonal variations in vegetation phenology, pest damage, and other biotic or abiotic stressors can significantly alter the spectral characteristics of both the LULC classes and the three coffee CS. These changes reduce their spectral separability and hinder accurate mapping when relying on a single image or spectral sampling from only one season. Therefore, future studies should incorporate multi-seasonal imagery and Ems derived from both dry and wet seasons, as well as from varying stress conditions, to enhance the robustness and reliability of classification accuracy”.

Comment 6:

Although the authors mention Robusta coffee productivity in the abstract and introduction, the study presents no results addressing it. The authors should either remove this content or provide additional data and analyses to demonstrate the study’s relevance to Robusta coffee productivity.

Authors’ response

Thank you for this observation. We agree and removed productivity from the abstract and introduction.

Reviewer #4: Overall Assessment

This manuscript presents an innovative two-step remote sensing approach for mapping Robusta coffee cropping systems (CS) in Uganda, integrating Random Forest classification of LULC with subpixel MESMA spectral unmixing. The combination of Sentinel-2 imagery with in-situ hyperspectral endmembers is novel for East African coffee systems and provides valuable applied insights for coffee management and climate adaptation.

The work is generally well designed and clearly presented. However, several methodological and interpretive issues require clarification, particularly regarding endmember representativeness, seasonal bias in spectral data, and the absence of comparisons with alternative classifiers or auxiliary predictors (e.g., climatic or topographic variables).

Addressing these points will considerably strengthen the manuscript.

Overall, this is a promising and publishable study, provided the authors undertake major revisions.

Authors’ response

We thank the reviewer for the encouraging and insightful comments. We appreciate the recognition of our study’s novelty and methodological contribution to mapping Robusta coffee systems in Uganda. In response, we have addressed all methodological and interpretive concerns, including clarifying endmember representativeness, accounting for potential seasonal bias, and discussing comparisons with alternative classifiers and auxiliary predictors. These revisions have substantially improved the quality of the manuscript.

Strengths

• High relevance: Uganda is a key Robusta producer, and mapping coffee systems has direct implications for adaptation to climate change and value-chain management (Bunn et al., 2019) https://cgspace.cgiar.org/bitstreams/bd1b904e-271c-4e50-ad35-55321bc02f5/download?utm_source=chatgpt.com]; Bukomeko et al., 2019) [https://doi.org/10.1007/s10457-017-0172-8].

• Novel methodology: The two-step design—Random Forest for LULC followed by MESMA at subpixel scale—demonstrates a creative approach rarely applied to coffee agroforestry mapping.

• Integration of field hyperspectral data: Using in-situ spectra to derive endmembers increases ecological realism and reduces reliance on generalized spectral libraries (Somers et al., 2011)[https://doi.org/10.1016/j.rse.2011.03.003].

• Strong classification results: The reported accuracies (93.5% LULC; 89.9% coffee CS) are notable given the highly heterogeneous smallholder landscape.

• Transparency and data access: The provision of open data resources supports reproducibility and potential reuse by other researchers.

Reviewer #4 – Detailed point-by-point response

Major Concerns

Comment 1:

• Endmember representativeness

o The analysis is based on three endmembers sampled during a single dry season. This may not fully capture the spectral variability introduced by phenology, pest damage, or stress conditions (Mutanga & Skidmore, 2004) [https://doi.org/10.1080/01431160310001654923].

o The authors should discuss how seasonal changes might bias classification and outline future improvements (e.g., inclusion of wet-season spectra).

Authors’ response

We thank the reviewer for this constructive criticism, which was alo raised by the first reviewer. We agree with both reviewers and included this shortcoming in the discussion chapter (lines 594–6005) “Seasonal variations in vegetation phenology, pest damage, and other biotic or abiotic stressors can significantly alter the spectral characteristics of both the LULC classes and the three coffee CS (73). These changes reduce their spectral separability and hinder accurate mapping when relying on a single image or spectral sampling from only one season. Therefore, future studies should incorporate multi-seasonal imagery and Ems derived from both dry and wet seasons, as well as from varying stress conditions, to enhance the robustness and reliability of classification accuracy”. The relevant references have also been added to the References section (lines 871–872).

Comment 2:

• Comparison with alternative methods

o MESMA is well established, but no comparison is made with alternative classifiers or subpixel approaches (e.g., SMA, SVM, or deep learning; Kussul et al., 2017) [https://doi.org/10.1109/LGRS.2017.2681128].

o Even if not implemented, the rationale for selecting MESMA over newer approaches should be expanded.

Authors’ response

We thank the reviewer, particularly for this comment and for referring us to such an interesting paper by Kussul et al., (66). The rationale of selecting MESMA is expanded as suggested by the reviewer (Furthermore, unlike the spectral mixture analysis, which uses a fixed set of endmembers for all pixels, MESMA allows EM combinations to vary for each pixel, making it more adaptable to spatially heterogeneous surfaces. While deep learning methods can also capture complex spectral–spatial relationships, they typically require large datasets and high computational resources. MESMA provides a more transparent, data-efficient, and interpretable approach for mapping mixed land-cover types, especially in areas with limited training data or where physical understanding of spectral behavior is essential.), please see lines 95-101 in the introduction. We also added this argument “Although alternative subpixel classification approaches, such as support vector machine (SVM) (74) and deep learning (75), could have also been applied in this study, but a direct comparison between MESMA and these methods in mapping coffee CS was beyond this study scope. Future studies should explore the performance of different subpixel classifiers in mapping coffee CSs in the discussion (lines 525-532)

”.

Comment 3:

• Validation strategy

o The validation relies on polygons from the same period and region. This may overestimate accuracy (Tariq et al., 2023) [https://doi.org/10.1080/10095020.2022.2100287]. Consideration of independent temporal or spatial datasets would strengthen reliability.

Authors’ response

We thank the reviewer for this comment. We agreed and provided some discussion sentences. “Similarly, validation polygon samples were obtained from the same period and region to assess the performance of our classification approach. This approach, while practical, may lead to a slight overestimation of classification accuracy (59). Future studies should employ independent validation datasets collected across different time periods and geographic locations to enhance the reliability and robustness of the classification assessment.” in the discussion (lines 600-606)

Comment 4:

•Contextualization within broader agroecological debates

o The discussion could connect more explicitly with recent literature showing how heterogeneity in cropping systems influences remote sensing detectability and ecosystem outcomes (e.g., Escobar-López et al., 2022 [https://doi.org/10.3390/rs14163847]; Hunt et al., 2020) [https://doi.org/10.3390/rs12122041].

Authors’ response

We thank the reviewer for this comment. We addressed this in lines 546–551 “However, our study area represents a highly heterogeneous cropping system, where the structural and compositional diversity of Robusta coffee CSs influences both classification performance and ecological outcomes. This relationship aligns with broader agroecological evidence showing that variations in cropping system heterogeneity strongly affect remote sensing detectability as well as landscape-scale ecological functions (5, 13)”.

Comment 5:

o In particular, Cassamo et al. (2022)[https://doi.org/10.1016/j.agee.2022.108341] emphasize that mapping cropping systems provides a foundation for assessing ecosystem services and sustainability in smallholder agroecosystems. This perspective complements the current work and underscores its potential applications beyond classification, for example in policy design and environmental assessments.

Authors’ response

We thank the reviewer for this valuable suggestion. In lines 491–499, we have expanded the discussion to emphasize the broader relevance of our findings, noting, as highlighted by Cassamo et al. (2022), that mapping cropping systems provides a foundation for assessing ecosystem services, sustainability, and policy applications “Importantly, mapping Robusta coffee CSs extends beyond simple land-cover classification. As noted by Cassamo et al. (70), detailed characterization of cropping systems forms the foundation for assessing ecosystem services and sustainability within smallholder agroecosystems. This highlights the broader policy and environmental relevance of our findings for coffee-producing regions, where spatially explicit insights can inform sustainable land management and climate-resilient agricultural planning.”

Minor Concerns

Comment 6:

• Methodological detail: Some processing steps (e.g., spectral resampling in hsdar, MESMA implementation in RStoolbox) could be described more explicitly for reproducibility.

Authors’ response

We have revised the processing stems of spectral resampling and MESMA for explicit description (lines 285-292).

Comment 7:

• Figures: Several figures (Figs. 5–7) would benefit from clearer legends and larger font size.

Authors’ response

We have addressed the comment by improving the clarity of the figures. Specifically, Fig. 7 has been split into two separate graphs for better readability: Fig. 9a now presents the frequency of pixel fraction values and threshold sensitivity, while Fig. 9b shows the spectral angle mapper (SAM) values, quantifying spectral separability. Additionally, we have increased the font size of axis titles, labels, and legends across all figures to enhance clarity.

Comment 8:

• Terminology consistency: Ensure consistent use of “coffee cropping systems (CS)” rather than alternating with “coffee systems.”

Authors’ response

Addressed, thank you.

Comment 9:

• References: Update with recent contributions on coffee/agroforestry mapping (Escobar-López et al., 2022; Kelley et al., 2018 [https://doi.org/10.3390/rs10060952]).

Authors’ response

We have cited all these references, thank you for referring us to such interesting papers, which we learned a lot from them.

Comment 10:

• Language polishing: Minor English corrections needed (e.g., “the Robusta CS class was masked” → “the Robusta coffee class was masked”).

Authors’ response

Thank you for pointing this out. However, “the Robusta coffee cropland class was masked” is the correct phrasing (see lines 28–29), and we have ensured consistency throughout the manuscript. We also polished the manuscript for other language error.

Comment 11:

The manuscript makes a strong methodological and applied contribution, but requires clarification of uncertainties (seasonality, validation limits), expanded discussion of alternatives, and broader contextualization with recent agroecological literature.

Importantly, the recommendation of “major changes” reflects the need for expanded analysis and framing, not a flaw in the current methodology. With revisions, the study can significantly advance understanding of coffee cropping system mapping in Africa

Authors’ response

We appreciate the positive assessment of our methodology and contribution. We have addressed the concerns by clarifying uncertainties (seasonality, validation limits), expanding the discussion of alternative approaches, with recent agroecological literature to strengthen and advance the overall framing understanding of coffee cropping system mapping in Africa.

---

## [Decision Letter · Decision Letter 2]

28 Nov 2025

Mapping Robusta coffee (Coffea canephora) cropping systems in Uganda: A two-step pixel and sub-pixel based approach with Sentinel-2 data

PONE-D-25-11912R2

Dear Dr. Aragaw,

We’re pleased to inform you that your manuscript has been judged scientifically suitable for publication and will be formally accepted for publication once it meets all outstanding technical requirements.

Kind regards,

Tzen-Yuh Chiang

Academic Editor

PLOS ONE

Additional Editor Comments (optional):

Reviewers' comments:

Reviewer's Responses to Questions

**Comments to the Author**

Reviewer #3: All comments have been addressed

2. Is the manuscript technically sound, and do the data support the conclusions?

Reviewer #3: (No Response)

3. Has the statistical analysis been performed appropriately and rigorously?

Reviewer #3: (No Response)

4. Have the authors made all data underlying the findings in their manuscript fully available?

Reviewer #3: (No Response)

5. Is the manuscript presented in an intelligible fashion and written in standard English?

Reviewer #3: (No Response)

Reviewer #3: (No Response)

**Do you want your identity to be public for this peer review?** For information about this choice, including consent withdrawal, please see our Privacy Policy

Reviewer #3: No

---

## [Editor Report · Acceptance letter]

PONE-D-25-11912R2

PLOS One

Dear Dr. Aragaw,

I'm pleased to inform you that your manuscript has been deemed suitable for publication in PLOS One. Congratulations! Your manuscript is now being handed over to our production team.

Kind regards,

on behalf of

Dr. Tzen-Yuh Chiang

Academic Editor

PLOS One